# CLeAR: Continual Learning on Algorithmic Reasoning for Human-like Intelligence

**Bong Gyun Kang**[1*]     **HyunGi Kim**[2]     **Dahuin Jung**[2]     **Sungroh Yoon**[1,2†]

[1] Interdisciplinary Program in Artificial Intelligence, Seoul National University
[2] Department of Electrical and Computer Engineering, Seoul National University

## Abstract

Continual learning (CL) aims to incrementally learn multiple tasks that are presented sequentially. The significance of CL lies not only in the practical importance but also in studying the learning mechanisms of humans who are excellent continual learners. While most research on CL has been done on structured data such as images, there is a lack of research on CL for abstract logical concepts such as counting, sorting, and arithmetic, which humans learn gradually over time in the real world. In this work, for the first time, we introduce novel algorithmic reasoning (AR) methodology for continual tasks of abstract concepts: CLeAR. Our methodology proposes a one-to-many mapping of input distribution to a shared mapping space, which allows the alignment of various tasks of different dimensions and shared semantics. Our tasks of abstract logical concepts, in the form of formal language, can be classified into Chomsky hierarchies based on their difficulty. In this study, we conducted extensive experiments consisting of 15 tasks with various levels of Chomsky hierarchy, ranging from in-hierarchy to inter-hierarchy scenarios. CLeAR not only achieved near zero forgetting but also improved accuracy during following tasks, a phenomenon known as backward transfer, while previous CL methods designed for image classification drastically failed.

## 1 Introduction

From an early age, humans develop their ability to solve complex logical reasoning problems through lifelong learning. They gradually learn sequential cognitive skills, from basic counting methods to more advanced concepts such as addition, subtraction, and logical operations [2]. Although this sequential learning ability is inherent in humans, deep learning models experience catastrophic forgetting [30], where their performance on previous tasks rapidly declines, after learning a new task. Continual learning (CL) [6, 19] is essential not only for practical purposes but also for developing human-like artificial intelligence that can maintain its performance on previous tasks after learning new tasks. However, existing CL research has been limited to learning streams of structured data with fixed input formats, such as images, where each task involves the same classification task only with different image classes [47, 1, 48]. Current approaches fail to reflect the learning of abstract logical concepts that humans continuously acquire in the real world.

There are three major challenges in naively leveraging existing CL methodologies to continual algorithmic reasoning (AR) tasks. Firstly, in AR tasks, input data contain little information, and the input data and task are decorrelated (Fig. 1b). In traditional image CL, the inherent information in the image itself enables the model to estimate which task the image belongs to [18]. However, in CL for abstract concepts, the input data may be completely decorrelated from the task that the model needs

---

*luckypanda@snu.ac.kr   • The code is available at https://github.com/Pusheen-cat/CLeAR_2023
†Corresponding author (sryoon@snu.ac.kr)

37th Conference on Neural Information Processing Systems (NeurIPS 2023).

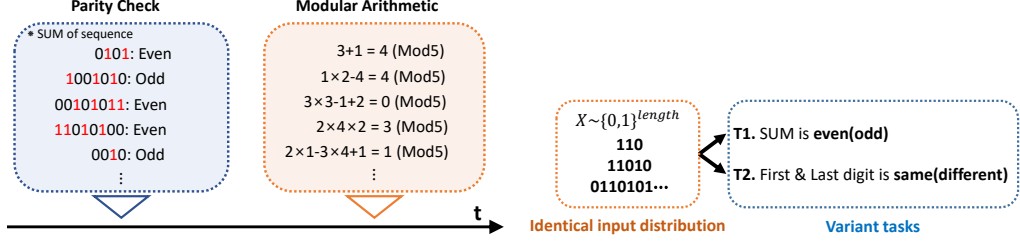

|                                 |                                |
| :-----------------------------: | :----------------------------: |
| (a) Sequential Tasks of Learning Logics | (b) Input-Task Decorrelation |

Figure 1: Introduction of continual learning on algorithmic reasoning (a) showing two representative continuous AR tasks and (b) showing an example of two tasks sharing identical input distribution.

to learn. For example, consider a binary input sequence consisting of 0's and 1's. In the first task, the model needs to determine whether the sum of the sequence is odd or even, while in the second task, it needs to determine the square root of the sequence when treated as a binary number. In this case, the input data distribution is identical for both tasks, thus it is impossible to correctly predict the task based on the input data. Therefore, CL for AR of abstract concepts is better suited to task incremental learning. Moreover, since the input data contains little information, learning data distribution (e.g. contrastive learning based CL) is inapplicable.

Secondly, the dimensions of the datasets of each task can be dynamic in the AR of abstract concepts. In current CL, data has mostly fixed shape and dimension (ex. height, weight, and channel $H \times W \times C$) throughout the tasks [47, 1, 48]. However, ways of representing each "abstract logical concept" as the sequential input-output format are various [11, 26]. For example, when trying to train the model for modular arithmetic, there are various ways to introduce the concepts of numbers and operators to the model. Specifically, all used numbers and operators can be one-hot encoded numbers, and operators can be represented in different dimensions, or numbers can be represented in binary format. Furthermore, input sequences of different tasks result in different dimensions and lengths. Therefore, it is impossible to apply the same model directly to each task's data like the traditional image CL methodology, and additional operations adjusting the data shape are necessary.

Thirdly, the goal of generalization that the AR task aims to achieve is different. In traditional CL, the model learns the input data distribution itself, assuming that the train dataset and test dataset are identical and independently distributed (IID) [45]. However, in AR tasks, the model learns the general rule that generates labels from input distribution. To identify whether the model has appropriately learned to generate the correct answers rather than simply memorizing them, the test set is composed of out-of-distribution (OOD) data [33] that was never seen during training. Therefore, the goal of AR is not generalization performance for IID as in traditional CL, but generalization for OOD [14]. Due to these characteristics, the performance of replay-based CL methods that use a replay buffer to represent the input data distribution decreases. The core of the replay buffer is to extract representative samples or prototypes that can represent the input data distribution [5]. However, unlike a sharp-eared and long-tailed image that represents a cat class, "5+8=13" cannot be considered as representative of the concept of arithmetic.

**This work.** We have developed a CL methodology that can be applied in the aforementioned scenario. We have applied a novel one-to-many mapping strategy to handle input data of different formats, and through this mapping, we have proposed ways to prevent forgetting previous tasks when learning new tasks, and even increased performance on previous tasks during the process of learning new tasks. We conducted extensive experiments on sequential models such as RNN, LSTM, and Tape RNN [12, 15, 17, 43, 11], which have memory, and used AR tasks with various levels of Chomsky hierarchy [11] to conduct extensive experiments on sequential tasks of the same level and different levels.

- We introduce the scenario of CL on AR for the first time. This scenario is of great importance as it is similar to how humans learn logical reasoning abilities incrementally in the real world and has very different characteristics from conventional CL scenarios as described above.

- We introduce new data mapping methods and CL algorithms that are applicable to the aforementioned scenario and have demonstrated high performance not only in catastrophic forgetting but also in knowledge transfer, which is one of the ultimate goals of CL.

- We demonstrated the superiority of our method over the existing CL methods in our proposed CL scenario for AR through extensive experiments on various combinations of AR tasks. We also showed that existing CL methods perform poorly due to the reasons described above.

## 2 Preliminary

**Continual learning.** CL is one of the long-standing challenges in deep learning [35], aiming to preserve the performance of previous tasks while effectively learning new tasks (without access to data from previous tasks) in situations where multiple tasks are learned sequentially. The existing CL methodologies can be broadly classified into three categories based on the methods used to prevent catastrophic forgetting [24]. First, regularization-based methods [19, 22, 21] constrain the model's important parameters for the previous task from changing when learning a new task. Elastic Weight Consolidation (EWC) [19] approximates the Fisher information matrix to learn a loss-minimizing weight space and regularizes the gradient direction to maintain the low loss for previous tasks. Learning without forgetting (LwF) [22] regularizes the loss through knowledge distillation by using the output of previous tasks as pseudo-labels during the learning of new tasks. Second, memory-based methods [5, 27, 4, 37] are further divided into experiment-replay and orthogonal projection methods. Experiment replay (ER) [5] stores small amounts of data from previous tasks in memory to use their information when learning new tasks. The performance of this method depends on the representative sample selected to capture the entire distribution of data. Orthogonal projection methods [27, 4] stores gradient and update weight in a direction orthogonal to the gradient from the previous task. Third, parameter isolation-based methods [39, 28, 40] use different parts of the model's parameters for each task. The parameters used in previous task learning are masked during the learning of the next task, and the model can also be extended as new tasks are introduced.

**Knowledge Transfer.** Recent CL models aim not only to prevent catastrophic forgetting of previous tasks but also to enhance the overall performance of the entire task by transferring common knowledge between tasks [24]. This is a property that humans, as excellent continual learners, naturally possess, allowing them to learn new tasks more easily from previous experiences and gain a deeper understanding of previously acquired knowledge as they learn new tasks. However, such knowledge transfer is not easy for deep learning models to obtain. Backward transfer refers to improving the performance of previously learned tasks by learning the current task, while forward transfer refers to previously learned tasks helping to learn a new task. Although there have been studies in this area [39, 25], knowledge backward transfer is still hard to achieve [46].

**Algorithmic Reasoning.** The core of human logical reasoning ability lies in inductive inference [20, 36]. This is the ability to derive general rules from finite examples and use them to infer the correct answers from unseen data. From the perspective of deep learning, this is AR [11, 23], where the model receives a finite number of input-output data and infers the underlying general rule to generate output for input sequences that have not been seen before. There has been much research on program induction [34, 23, 32, 31, 11], however, conventional approaches for serial AR have a critical limitation in that they require prior human knowledge about the task to pre-configure the model [29, 44, 13]. In general deep learning, the model learns through empirical risk minimization of statistical learning theory to achieve the theoretical bound of its performance and estimates the generalization error using test data [45]. An important assumption here is that the train and test data are independent and identically distributed (IID). However, in AR, where the model needs to learn general rules, this assumption is violated [11]. Therefore, in AR tasks, we have to measure the model's performance not only on the train data distribution but also on out-of-distribution (OOD) distribution [11].

**Formal Language and Chomsky Hierarchy.** Computational problems that generate output through learnable rules can be represented in formal language [41, 8, 38]. Formal language can be classified into hierarchies based on their complexity, which is known as the Chomsky hierarchy [3, 9]. The hierarchy ranges from regular language, which requires the lowest complexity of memory, context-free language, which requires stack-type memory, context-sensitive language, which requires linear tape memory, and recursively enumerable language, which requires infinite tape memory. Research has been conducted on whether various deep learning models can learn each level of formal language in this hierarchy [11, 12, 7]. Results have shown that there is a Chomsky hierarchy bound on the formal language that each deep learning model can learn, whereas models with external memory can learn languages of a high Chomsky hierarchy.

# 3 Method

**Problem Statement.** Our goal is to train a fixed-size model that sequentially learns various abstract logical tasks, following a standard CL scenario. We assumed that predicting the output sequence corresponding to the input sequence, as in [11], is closer to real-world scenarios which are learned through inductive inference than inferring the internal state of automata, as in [26]. There are various methods of representing the same abstract logical concept into an input sequence that a model can understand. However, different forms of input can affect the performance of the model. Therefore, for fairness, we used the data format proposed in the previous paper [11].

In a CL scenario, a sequence of tasks $\mathbb{T} = \{\text{task } t\}_{t=1}^{T}$ is given one after another. Each task $t$ is given as a dataset $\mathbb{D}_t = \{(\mathbf{x}_{t,i}, \mathbf{y}_{t,i})\}_{i=1}^{S_t}$ with $S_t$ sample pairs. Each pair of $\mathbf{x}_{t,i}, \mathbf{y}_{t,i}$ comes from $\left\{ \text{One-hot}(L_{in}^t) \in \{0,1\}^{(|\sum_{in}^t|, l_{in} \leq N)}, \text{One-hot}(L_{out}^t) \in \{0,1\}^{(|\sum_{out}^t|, l_{out})} \right\}$, where $L_{in}^t \in \left\{ \sum_{in}^t \right\}^{l_{in} \leq N}$ and $L_{out}^t \in \left\{ \sum_{out}^t \right\}^{l_{out}}$ is input and output language pair from task $t$. $N$ is given as the input sequence length limit. $\sum_{in}^t$ and $\sum_{out}^t$ are corresponding alphabets.

Consider a sequence-to-sequence model $f_\theta(\cdot)$ parameterized by fixed-sized $\theta$. To match different sized input $\mathbf{x}_{t,i}$ of each task to model input, we require a task-wise mapping function $m_t(\cdot)$. Also, to project the model output vector to match different sized output $\mathbf{y}_{t,i}$, we applied task-wise linear projection head $h_t(\cdot)$ which consists of a single layer perceptron. After learning a $k$-th task, the model is parameterized by $\theta(k)$, and the projection head becomes $h_{t,k}(\cdot)$.

During training, the model can only access data from the current task. And after training, to evaluate whether the model actually learned the general output sequence generation rule, the model is tested on OOD dataset $\mathbb{D}_t^{test} = \{(\mathbf{x}_{t,i}^{test}, \mathbf{y}_{t,i}^{test})\}_{i=1}^{S_t^{test}}$ which are created in the same method as train dataset from $L_{in}^t \in \left\{ \sum_{in}^t \right\}^{N < l_{in} \leq M}$ and its corresponding output sequence $L_{out}^t \in \left\{ \sum_{out}^t \right\}^{l_{out}}$. After learning a $k$-th task, model performance on task $t$ is measured using an average of per-sequence accuracy of test data, i.e., $\text{ACC}_{k,t}^{test}$.

$$\text{ACC}_{k,t}^{test} := \frac{1}{S_t^{test}} \sum_i A_{t,i}^k(\boldsymbol{x}, \boldsymbol{y})$$

$$A_{t,i}^k(\boldsymbol{x}, \boldsymbol{y}) := \frac{1}{|\boldsymbol{y}|} \sum_l \mathbb{I}\left[ (\text{argmax}_j \cdot \boldsymbol{y}_{jl}) = (\text{argmax}_j \cdot h_t(p_{\theta(k)}(m_t(\boldsymbol{x})))_{jl}) \right]$$

Following [11], we set the maximum training input sequence length to be 40 ($N = 40$) and sampled input sequence length from the uniform distribution $\mathcal{U}(1, N)$. OOD test set input sequence length was sampled from the uniform distribution $\mathcal{U}(N + 1, M)$ where $N = 40$ and $M = 100$.

**Tasks.** We conducted a comprehensive evaluation of CL on AR (CL-AR), involving 15 diverse logical reasoning tasks. These tasks exhibit various levels of complexity, resulting in different Chomsky hierarchies. Table 1 shows our AR tasks and the corresponding Chomsky hierarchy. The detailed information for each task is in Appendix A. We conducted three major CL scenarios. (a) High-correlation CL-AR scenario continually learns the same kind of task with varying difficulty. One example is the Modular Arithmetic task with increasing modular value enabling the model to access larger numbers. Another example is the Cycle Navigation task with increasing cycle length. This scenario allows us to interpret the model's behavior upon learning highly correlated reasoning tasks. (b) In-hierarchy CL-AR scenario continually learns various tasks of the same level of Chomsky hierarchy. This scenario allows us to analyze the interaction among tasks of different natures but with similar complexity. (c) Inter-hierarchy CL-AR scenario consists of tasks of different Chomsky hierarchies. This is by far the most challenging scenario for a neural network that has to manipulate its memory structure.

**Models.** Recently, there have been numerous quantitative studies on whether various architectures of neural networks can learn formal languages. In recent research, it has been shown that regardless of the increase in the number of training data, models have "hard limitations for scaling laws" and cannot learn generalization rules based on the complexity of the task and the inductive bias of the model. Furthermore, to learn high-level tasks, models with external memory are necessary, and the commonly used transformer model has demonstrated very limited AR performance. Therefore, in

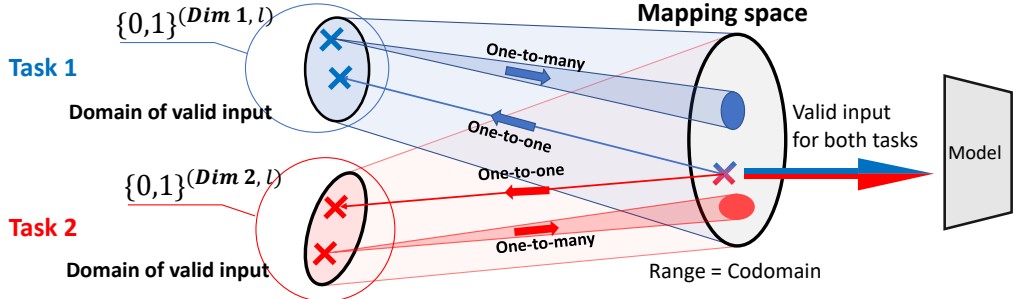

Figure 2: $m_t$ maps this input space to the entire mapping space, which is shared across all tasks. In order to map the entire space, mapping from the input space to the mapping space become one-to-many, making an arbitrary point in mapping space to be mapped from a valid input sequence for every task. Also, this mapping function ensures that the distribution of inputs from the input space becomes a uniform distribution in the mapping space.

this paper, we used RNN [12] and LSTM [15], as well as stack RNN [17] and tape RNN [43, 11] which have additional differentiable memory structures added to the standard RNN. The length of the external memory stack and tape in stack RNN and tape RNN was set to 40, and each element had 8 dimensions [11] and dimensions were doubled with CL of more than 5 tasks.

**Evaluation Metrics.** We adopt generalization accuracy as the basic evaluation metric. Final model accuracy (ACC) is the average accuracy of all tasks after learning all tasks as follows:

$$\text{ACC} = \frac{1}{T} \sum_{t=1}^{T} \text{ACC}_{T,t}^{test} \tag{1}$$

In order to evaluate not only the catastrophic forgetting but also the model's knowledge transfer capability, we introduced a metric that measures backward/forward knowledge transfer [42, 16]. Backward knowledge transfer (BWT) [27] indicates how much learning on the next tasks has influenced the performance of the previous task. BWT $> 0$ indicates that there has been a transfer of beneficial knowledge to the previous task during the CL process. Forward knowledge transfer (FWT) measures how helpful previously learned knowledge is when learning a new task. FWT is obtained by calculating the difference between the accuracy achieved in the CL scenario and the accuracy achieved by training an independent model (as Table 1). BWT and FWT are expressed as follows:

$$\text{BWT} = \frac{1}{T-1} \sum_{t=1}^{T-1} \text{ACC}_{T,t}^{test} - \text{ACC}_{t,t}^{test}, \quad \text{FWT} = \frac{1}{T-1} \sum_{t=1}^{T-1} \text{ACC}_{t,t}^{test} - \text{ACC}_{t}^{test} \tag{2}$$

**CLeAR.** We propose a novel CL algorithm for the CL-AR scenario: CLeAR. The CLeAR model is divided into two main parts. The first part is a mapping function $m_t$ that maps sequences sampled from the input data distribution to a shared mapping space. The second part consists of the main model $f_\theta(\cdot)$, which learns labels from the features in the feature space, and the task-wise single-layer projection head $h_t$.

$m_t$ maps the input sequence sampled from the input space of task $t$ to a feature space (mapping space), preserving length but changing dimension to fixed size $D_{map}$. The input sequence consists of binary values, denoted as $\{0,1\}^{(D_{in}^t, l)}$, and the range of the mapping space is expressed as $\{0,1\}^{(D_{map}, l)}$. As described in Fig. 2,

---

**Algorithm 1 CLeAR**-training procedure for a given task t

**Input:**
Trained up to task t-1: $m_k$, $f_{\theta(t-1)}$ and $h_{k,t-1}$ $(1 \leq k < t)$
Dataset of task t: $\mathfrak{D}_t$
**Train:**
Train $m_t$ with $\mathfrak{D}_t$
$f_\theta = f_{\theta(t-1)}$ and $h_k = h_{k,t-1}$ $(1 \leq k < t)$
Freeze $f_{\theta(t-1)}$, $h_{k,t-1}$ $(1 \leq k < t)$
Random initialize $h_t$
**for** $x_{t,i}, y_{t,i} \sim \mathfrak{D}_t$ **do** {*Pre-tuning with small samples*}
    $o_{t,i} \leftarrow h_t \cdot f_{\theta(t-1)} \cdot m_t(\mathbf{x}_{t,i})$
    $\mathcal{L} = \mathcal{L}_{\text{new}}(o_{t,i}, y_{t,i})$
    Compute $\nabla \mathcal{L}$ and update $h_t$
**end for**
**for** $x_{t,i}, y_{t,i} \sim \mathfrak{D}_t$ **do** {*Main training*}
    $\hat{y}_{k,i} \leftarrow argmax \cdot h_{k,t-1} \cdot f_{\theta(t-1)} \cdot m_t(\mathbf{x}_{t,i})$ for $k = 1 \cdots t-1$

    $o_{k,i} \leftarrow h_k \cdot f_\theta \cdot m_t(\mathbf{x}_{t,i})$ for $k = 1 \cdots t$
    $\mathcal{L} = \lambda \cdot \sum_{k=1}^{t-1} \mathcal{L}_{\text{old}}(o_{k,i}, \hat{y}_{k,i}) + \mathcal{L}_{\text{new}}(o_{t,i}, y_{k,i})$
    Compute $\nabla \mathcal{L}$ and update $f_\theta, h_k$ for $k = 1 \cdots t$
**end for**
**Output:**
Trained up to task t: $m_k$, $f_\theta$ and $h_k$ $(1 \leq k \leq t)$

---

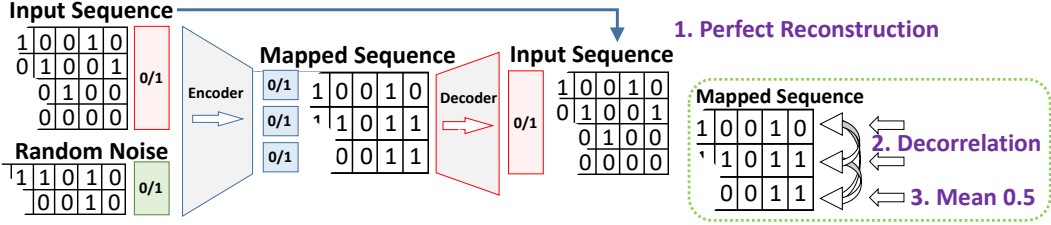

Figure 3: Mapping function is trained using encoder-decoder architecture and under three constraints. First, input sequence is perfectly reconstructed. Second, each column of mapped sequence is decorrelated. Third, each column of mapped sequence has mean of 0.5.

our mapping function $m_t$ has the following characteristics. First, an arbitrary sample from the input distribution is mapped to one or more points in the mapping space (one-to-many). Second, for any point in the mapping space, a corresponding sample exists in the input space. Third, the mapping of input distribution resembles uniform sampling in the mapping space: $m_t(input\,distribution) \approx \mathcal{U}(\{0,1\}^{(D_{map},l)})$. Therefore, the mapping function of each task expands the input distribution to fill the mapping space fully. Thus, an arbitrary sample drawn from the mapping space can correspond to a valid input sample for any task. This allows the model to consider a single sequence in the mapping space as valid input for each previous task. It is similar to how we can perceive the number 395 in various ways, such as being odd, multiple of 5, or having a sum of digits equal to 17. These mapping functions can be expressed very simply, requiring minimal computational resources in the overall model. Additional details about the mapping function, positional mapping, and specific mapping for each task are explained in Appendix C.

To create such mapping functions, an encoder-decoder structure is considered for tasks where the distribution of alphabets is preserved across all positions of the input language. As shown in Fig. 3, the encoder takes the input sequence sampled from the input space and additional random noise to generate a mapped sequence. Then, the decoder reconstructs the input sequence from the mapped sequence. In this process, each column of the mapped sequence should have a mean of 0.5 and should be decorrelated from each other. The encoder serves as the mapping function, and the decoder is discarded.

The functions $f_\theta(\cdot)$ and $h_t(\cdot)$ are trained using a similar approach to LwF [22]. After completing the training for the $t$-1 th task, the training proceeds for the $t$ th task, temporarily saving $f_{\theta(t-1)}$ and $h_{k,t-1}$ for this task. First, $m_t$ is trained. Then, the randomly initialized $h_t$ is pre-tuned while the rest of the model is frozen using a small portion of the dataset. For each sample $\mathbf{x}_{t,i}, \mathbf{y}_{t,i}$, the pseudo-label for the $k$-th ($1 \leq k < t$) task $\hat{y}_{k,i}$ is obtained by $argmax \cdot h_{k,t-1} \cdot f_{\theta(t-1)} \cdot m_t(\mathbf{x}_{t,i})$. The loss is calculated for all $t$ heads. For the outputs of the $k$-th task heads: $h_k \cdot f_\theta \cdot m_t(\mathbf{x}_{t,i})$ ($1 \leq k \leq t$), the loss is computed with the pseudo-labels $\hat{y}_{k,i}$ for $1 \leq k < t$, and for the $t$-th task head, the loss is computed with $\mathbf{y}_{t,i}$. The final loss is the sum of the losses for each head up to $t$-th. The overall learning process is shown in Algorithm 1.

## 4 Experiments

In this section, we first confirmed that applying our mapping function and tuning structure did not alter the learning ability of each model on tasks from various Chomsky hierarchies, compared to the previous study [11]. Next, we demonstrate CLeAR's superior performance on high-correlation, in-hierarchy, and inter-hierarchy CL-AR tasks through extensive ablations. Furthermore, we analyze the model's robustness in terms of noise within the mapping space. A detailed explanation of the experiment setting and comparison with existing CL algorithms [19, 22, 5] and their failure is shown in Appendix D.

### 4.1 Baseline Generalization Accuracy

In Table 1, we presented the generalization accuracy of each model for all tasks in our paper. Corresponding in-distribution accuracy and reported generalization accuracy in the previous paper [11] paper is shown in Appendix B. It was shown that applying mapping and head did not greatly affect the model's performance and did not alter the limits of the Chomsky hierarchy that each model can learn. The reported performance here was used as a benchmark for calculating Single-task accuracy and FWT.

Table 1: Baseline generalization accuracy (%) of overall tasks. Each task was averaged on three repeats, trained for 50,000 epochs each. The accuracy value was reported with positional mapping (accuracy without positional mapping is in parentheses). For accuracy over 90%, we considered that the model successfully learned AR as suggested in [11]. The random accuracy is 50% except for Modular Arithmetics (R), Cyclic Navigation (R), and Bucket Sort (CS). Tasks that require additional positional mapping are tagged with $^\dagger$.

| Level | Task | RNN | Stack-RNN | Tape-RNN | LSTM |
|-------|------|-----|-----------|----------|------|
| Regular | Even Pairs | **100.0**(100.0) | **100.0**(100.0) | 51.4(53.1) | **100.0**(100.0) |
| | Modular Arithmetic | **99.7**(98.8) | **99.1**(99.2) | 63.1(82.4) | **100.0**(100.0) |
| | Parity Check | **100.0**(100.0) | **100.0**(100.0) | **99.6**(99.3) | **100.0**(100.0) |
| | Cycle Navigation | **100.0**(98.1) | **99.7**(100.0) | 80.0(97.3) | **92.1**(87.9) |
| Context Free | Compare Occurrence | **96.6**(97.9) | **99.5**(98.5) | **97.7**(97.9) | **99.6**(99.5) |
| | Stack Manipulation$^\dagger$ | 57.0 | 83.9 | 75.6 | 82.4 |
| | Reverse String | 77.7(79.9) | 80.1(80.0) | 51.3(56.2) | 79.3(79.8) |
| | Divide by 2$^\dagger$ | 71.4 | **92.0** | 59.8 | 70.0 |
| Context Sensitive | Duplicate String | 54.3(53.2) | 58.0(60.1) | 51.3(51.6) | 74.2(76.9) |
| | Interlocked Pairing$^\dagger$ | 60.6 | **96.7** | 53.8 | **99.7** |
| | Odds First | 56.5(56.5) | 60.8(61.3) | 61.4(61.4) | 73.2(72.3) |
| | Binary Addition$^\dagger$ | 48.6 | 55.4 | 51.9 | 71.3 |
| | Binary Multiplication$^\dagger$ | 50.1 | 52.3 | 48.9 | 60.9 |
| | Compute Sqrt | 57.9(57.4) | 64.4(64.1) | 56.3(60.3) | 66.3(66.7) |
| | Bucket Sort | 59.3(55.9) | **96.7**(97.5) | 71.4(78.7) | **97.2**(98.7) |

Table 2: Generalization accuracy (%) on high-correlation CL-AR scenario, averaged on three repeats. Joint refers to simultaneously training all tasks, with tasks alternating at each step and ultimately being learned uniformly. Single refers to averaging the accuracy of each individual task. CL Initial averages the accuracy immediately after learning each task, while CL Final is the final average accuracy. CL Final values that are greater than 90% or the Joint accuracy were marked in bold, as well as the positive BWT values.

| Task | Model | Average Accuracy (%) | | | | |
|------|-------|-------|--------|------------|----------|------|
| | | Joint | Single | CL Initial | CL Final | BWT |
| Modular Arithmetic | RNN | 40.32 | 99.70 | 61.08 | **59.68** | -1.64 |
| | Stack-RNN | 54.24 | 97.98 | 63.27 | **62.66** | -0.72 |
| | Tape-RNN | 43.68 | 84.72 | 30.84 | 24.58 | -7.31 |
| | LSTM | 93.07 | 99.96 | 96.38 | **96.25** | -0.15 |
| Cycle Navigation | RNN | 96.33 | 99.36 | 98.57 | **98.09** | -0.56 |
| | Stack-RNN | 96.97 | 99.83 | 98.52 | **97.88** | -0.75 |
| | Tape-RNN | 56.93 | 88.65 | 65.46 | **67.76** | **2.67** |
| | LSTM | 98.72 | 96.18 | 99.54 | **99.37** | -0.20 |
| Reverse String | RNN | 46.46 | 64.11 | 42.64 | 40.42 | -2.60 |
| | Stack-RNN | 60.00 | 67.84 | 68.39 | **67.69** | -0.81 |
| | Tape-RNN | 24.77 | 40.53 | 42.93 | **39.48** | -4.02 |
| | LSTM | 46.35 | 65.98 | 50.73 | **48.91** | -2.12 |
| Bucket Sort | RNN | 51.43 | 56.06 | 54.25 | 51.36 | -3.36 |
| | Stack-RNN | 93.49 | 94.19 | 95.16 | **92.44** | -3.18 |
| | Tape-RNN | 73.28 | 74.54 | 79.22 | **77.54** | -1.96 |
| | LSTM | 99.92 | 98.40 | 99.55 | **99.97** | **0.48** |

## 4.2 CLeAR on CL-AR

In this section, we demonstrate that the CLeAR can be applied to various abstract logical tasks, regardless of the model type and difficulty level. **Firstly**, we conducted experiments on a scenario where a single task becomes progressively more challenging (high-correlation scenario). Since the sequentially presented tasks require similar logical reasoning, there is a significant level of correlation between these tasks. For example, in the same Cycle Navigation task, the difficulty increases as the cycle size expands, while in Modular Arithmetic, the difficulty increases as the modular

Table 3: Generalization accuracy (%) on in-hierarchy CL-AR scenario, averaged on three repeats. Each task is abbreviated to: Cycle Navigation(CN), Modular Arithmetic(MA), Parity Check(PC), Even Pairs(EP), Divide by 2(DT), Reverse String(RS), Stack Manipulation(SM), Compare Occurrence(CO), Binary Addition(BA), Binary Multiplication(BM), Bucket Sort(BS), Compute Sqrt(CS), Duplicate String(DS), Interlocked Pairing(IP), Odds First(OF).

| Task | Model | Average Accuracy | | | | | |
| | | Joint | Single | CL Initial | CL Final | BWT | FWT |
|---|---|---|---|---|---|---|---|
| Regular CN-MA-PC-EP w/o CLeAR | RNN | 97.61 | 99.22 | 99.85 | 47.50 | -69.80 | - |
| | Stack-RNN | 96.50 | 99.79 | 99.42 | 47.64 | -69.04 | - |
| | Tape-RNN | 71.05 | 83.01 | 98.24 | 47.44 | -67.73 | - |
| | LSTM | 98.73 | 96.96 | 98.02 | 50.65 | -63.15 | - |
| Regular EP-PC-MA-CN w/o CLeAR | RNN | 97.79 | 99.22 | 98.72 | 53.80 | -59.90 | - |
| | Stack-RNN | 96.50 | 99.79 | 99.97 | 55.11 | -59.82 | - |
| | Tape-RNN | 71.05 | 83.01 | 68.24 | 47.56 | -27.58 | - |
| | LSTM | 98.73 | 96.96 | 97.48 | 53.61 | -58.48 | - |
| Regular CN-MA-PC-EP | RNN | 97.61 | 99.22 | 96.45 | **96.17** | -0.37 | -4.39 |
| | Stack-RNN | 96.50 | 99.79 | 94.89 | 87.09 | -10.40 | -10.47 |
| | Tape-RNN | 71.05 | 83.01 | 76.31 | **76.70** | **0.53** | -8.43 |
| | LSTM | 98.73 | 96.96 | 98.21 | **96.49** | -2.29 | 0.97 |
| Regular EP-PC-MA-CN | RNN | 97.79 | 99.22 | 99.68 | **99.80** | **0.16** | **0.72** |
| | Stack-RNN | 95.22 | 99.79 | 98.19 | **99.28** | **1.45** | -0.50 |
| | Tape-RNN | 59.61 | 83.01 | 58.16 | **74.45** | **21.72** | -27.17 |
| | LSTM | 98.72 | 96.96 | 99.99 | **99.96** | -0.05 | **4.01** |
| Context Free DT-RS-SM-CO | RNN | 65.95 | 75.66 | 64.13 | 61.67 | -3.27 | -14.13 |
| | Stack-RNN | 69.64 | 88.88 | 79.14 | **69.95** | -12.26 | -17.63 |
| | Tape-RNN | 54.06 | 71.09 | 62.26 | **62.07** | -0.25 | -9.77 |
| | LSTM | 82.72 | 82.86 | 76.93 | 79.36 | **3.24** | -4.57 |
| Context Free CO-SM-RS-DT | RNN | 65.95 | 75.66 | 64.87 | 63.89 | -1.30 | -14.96 |
| | Stack-RNN | 69.64 | 88.88 | 69.74 | 67.59 | -2.86 | -26.85 |
| | Tape-RNN | 54.06 | 71.09 | 74.42 | **67.67** | -9.01 | **0.63** |
| | LSTM | 82.72 | 82.86 | 75.69 | 74.78 | -1.21 | -9.14 |
| Context Sensitive BA-BM-BS-CS-DS-IP-OF | RNN | 55.33 | 55.33 | 50.77 | 50.51 | -0.34 | -5.28 |
| | Stack-RNN | 52.70 | 69.19 | 61.06 | **60.52** | -0.72 | -9.53 |
| | Tape-RNN | 45.97 | 56.43 | 46.77 | **45.97** | -1.07 | -12.60 |
| | LSTM | 76.73 | 77.54 | 73.48 | 74.11 | **0.84** | -3.38 |

value grows. Table 2 shows that CLeAR demonstrated very low forgetting for the most tasks, and in some cases, backward knowledge transfer occurred. In general CL, Joint is considered a soft upper bound [10]. We have highlighted CL Final when it exceeds that of Joint, as well as when it surpasses 90%, indicating successful learning of all tasks. Notably, CLeAR effectively preserve the best information they have learned even in cases where a task is not fully learned (low CL Initial) or when learning a specific task is deemed impossible for such a model [11]. **Secondly**, we conducted experiments on distinct tasks that belong to the same level of the Chomsky hierarchy (in-hierarchy scenario). This represents scenarios where the tasks are of different types but have similar difficulty levels. As evident from the first two rows of Table 3, simple sequential learning without using CLeAR completely forgets the previous task while learning the next task. In contrast, CLeAR exhibits low levels of forgetting for tasks across all Chomsky hierarchies and demonstrates multiple instances of backward/forward knowledge transfer (Table 3). Note that even in cases not highlighted in bold, CLeAR exhibited very little forgetting or near Joint performance. **Lastly**, we explored CL for tasks belonging to different levels of the Chomsky hierarchy (inter-hierarchy scenario). This entails tasks that differ both in type and difficulty level. Table 4 shows both RNN and LSTM are capable of storing information on tasks of different levels. During the learning process, which includes relatively long and completely unrelated tasks, both models successfully remember all the tasks learned from the very beginning.

Table 4: For inter-hierarchy CL-AR scenario, average generalization accuracy (%) is obtained for each group of tasks, averaged on three repeated experiments. The scenario consisted of ten AR tasks that followed an ascending order in the Chomsky hierarchy: EP-CN-PC-CO-RS-SM-IP-OF-BS-DS (abbreviations same with Table 3)

| Model | Chomsky hierarchy | Average Accuracy (%) | | | | |
|---|---|---|---|---|---|---|
| | | Joint | Single | CL Initial | CL Final | BWT |
| RNN | Regular | 60.50 | 100.0 | 93.31 | **75.04** | -27.40 |
| | Context Free | 59.94 | 77.10 | 61.59 | **60.78** | -0.81 |
| | Context Sensitive | 49.48 | 57.68 | 47.13 | 46.99 | -0.19 |
| LSTM | Regular | 96.76 | 97.37 | 100.0 | **93.46** | -9.06 |
| | Context Free | 79.70 | 87.10 | 80.94 | 79.11 | -1.83 |
| | Context Sensitive | 83.38 | 86.08 | 79.60 | 77.67 | -2.57 |

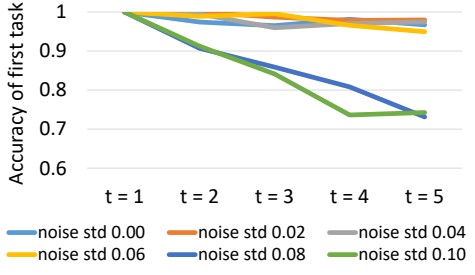

(a) Robustness against Different Noise Level

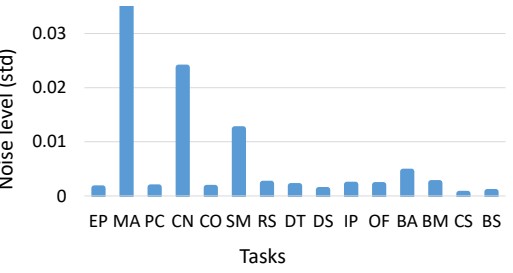

(b) Visualization of Noise Value of Each Task

Figure 4: An analysis of the impact of noise in the mapping space on CLeAR and an assessment of the amount of noise present in each task. In (a), we show that the proposed framework is robust enough till the noise level of std 0.06 for five continual tasks. In (b), we provide the noise level of the task's mapping space distribution for all 15 tasks and show they are all smaller than the notable noise level.

## 4.3 Robustness to Distribution Shift

Although we aim to map the input distribution to become uniformly distributed, this is not always possible. We provide a quantitative analysis of whether the model can maintain high CL performance without catastrophic forgetting, even in situations where the mapping distribution is not uniform. We conducted experiments by adding Gaussian noise with specific standard deviation (std) to the uniform probability distribution on the model's mapping space. As shown in Fig. 4(a), the model exhibited robustness even in the presence of significant noise (std up to 0.06 induces forgetting of 5% or less for the five continual tasks). For our mapping strategy, the std of mapping space distribution is much smaller than the notable noise level for all 15 tasks as shown in Fig. 4(b). This indicates that CLeAR performs well for task mappings with distributions that deviate to some extent from a uniform distribution, demonstrating high robustness.

## 4.4 Joint training of AR tasks

We discovered from the experimental results in Tables 2, 3 and 4 that continual learning of tasks for CL-AR often showed higher performance compared to when all tasks were learned simultaneously (Joint). This contrasts with the conventional view of continual learning that regards joint training accuracy as a soft upper bound [10]. Therefore, we analyzed the characteristics of the model in continual learning and simultaneous learning. First, we visualized the model's internal feature space for the simplest two tasks, `Parity Check` (PC) and `Even Pairs` (EP), as described in this paper [11], to illustrate how the model's learning occurs in each scenario in Fig. 5. From all the plots, we can observe that after the model learns PC or EP, it internally learns the corresponding automata for each task. Furthermore, after CL or joint training, it's evident that the hidden feature is clustered to solve both tasks simultaneously (Fig. 5 C, D). When learning EP following PC, the distribution of the hidden features remains similar, and it exhibits a pattern where each cluster further splits. However, in the case of joint training, the hidden features occupy a broader space, and unlike CL, the states for PC are clustered into eight pieces.

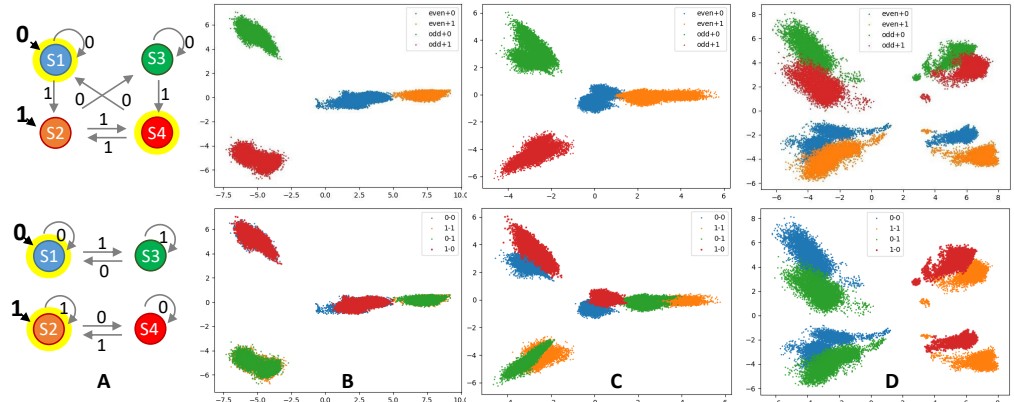

Figure 5: The **A** represents the automata for Parity Check (PC, up) and Even Pairs (EP, down). Each state (S1-S4) is color-coded to match the features of B, C, and D, with the final states for positive labels highlighted in yellow. Plots show the distribution of the model's hidden features using PCA visualization: after PC training (**B**), after PC-EP CL (**C**), and after PC-EP joint training (**D**). The upper row corresponds to PC automata states, and the lower row corresponds to EP automata states.

The difference between the two learning methods can be even more clearly observed in the following experiment. We separated the features of each model, trained with CL and joint training, based on the two consecutive digits in Fig. 6. These consecutive two digits are not necessary for learning PC or EP but represent easy-to-learn patterns that the model can easily grasp. In the case of joint training, this pattern is learned, and the feature is well separated, while in the case of

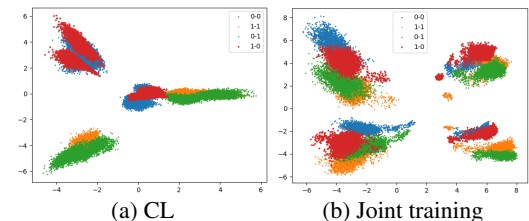

(a) CL          (b) Joint training

Figure 6: Clustering on two consecutive digits

CL, this information is not learned. These observations suggest that when learning tasks sequentially through CLeAR, previously learned tasks can act as a regularizer, reducing the acquisition of unnecessary information for the task.

## 5 Conclusion

Our study elucidates the distinct features of CL-AR as compared to conventional image-based CL and highlights the limitations of traditional CL approaches. Additionally, we introduce CLeAR, a novel methodology for the continual learning of abstract logical concepts. Experiments on 15 tasks with varying difficulty validate the efficacy of CLeAR. It achieves near-zero forgetting and often improves accuracy on previous tasks during subsequent training. There are major challenges specific to CL-AR tasks: the decorrelation between input data and tasks, dynamic dimensions of datasets, and the goal of generalization for OOD data. CLeAR tackles these challenges by employing a one-to-many mapping strategy that aligns tasks of different dimensions and shares mapping space. This prevents forgetting previous tasks and enhances overall performance. Our research contributes to the development of lifelong learning abilities, resembling human cognitive processes. By overcoming the limitations of existing CL approaches and focusing on abstract concepts, CLeAR opens new possibilities for continual learning. The limitation of our approach is that the mapping strategy used in our method involves discretized mapping, making it challenging to train the accompanying model end-to-end.

## 6 Acknowledgement

This work was supported by Electronics and Telecommunications Research Institute (ETRI) grant funded by the Korean government. [23ZS1100, Core Technology Research for Self-Improving Integrated Artificial Intelligence System], Institute of Information & Communications Technology Planning & Evaluation (IITP) grant funded by the Korea government (MSIT) [2021-0-01343: Artificial Intelligence Graduate School Program (Seoul National Univ.) and 2021-0-02068: Artificial Intelligence Innovation Hub], National Research Foundation of Korea (NRF) grant funded by MSIT (2022R1A3B1077720 and 2022R1A5A708390811) and the BK21 FOUR program of the Education and Research Program for Future ICT Pioneers, Seoul National Univ. in 2023.

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
