# A  Details on Tasks and Experiments

## A.1  Details on tasks

To prevent bias and ensure a fair comparison, all tasks were defined using the previously suggested format [6] of the input-output sequence generated from corresponding formal language sequences. Only for `Bucket Sort`, we modified the default number of digits of the input sequence from 5 to 4 for simplicity and mapping convenience. An intuitive description of each task is in C.2. Detailed descriptions of the following tasks: `Even Pairs`, `Modular Arithmetic`, `Parity Check`, `Cycle Navigation`, `Stack Manipulation`, `Reverse String`, `Duplicate String`, `Odds First`, `Binary Addition`, `Binary Multiplication`, `Compute Sqrt`, and `Bucket Sort` can be found in Appendix A of [6] and for these tasks: `Compare Occurrence`, `Divide by 2`, `Interlocked Pairing`, please refer to the previous version of [6] (https://arxiv.org/pdf/2207.02098v1.pdf).

## A.2  Details on experiments

To process the mapped input sequence, we used four types of sequential models employed in the aforementioned paper [6]: RNN, LSTM, Stack-RNN, and Tape-RNN. All models were trained in a non-autoregressive setting. Unlike the scenario of the previous paper, where one task was trained at a time, the nature of the continual learning setting requires a larger model capacity. Therefore, in the continual learning scenario for more than five tasks, we increased the hidden dimension of the model from 256 to 512. For Stack-RNN and Tape-RNN, we doubled the dimension of the memory elements ($R^d$) from $d = 8$ to $d = 16$. Also, instead of training 1,000,000 steps [6], we trained each task for 50,000 steps for every CL scenario. We used Adam optimizer [7] with a learning rate $1.0 \times 10^{-3}$ and applied gradient clipping with a threshold of 1.0. We used a batch size of 128 for training. All experiments were conducted using the Jax framework [2, 1]. RNN, LSTM, Stack-RNN, and Tape-RNN codes are based on [6]. Each scenario was trained using one NVIDIA GeForce RTX 3090Ti GPU or NVIDIA A40 GPU. The code is available in our git repository (https://github.com/Pusheen-cat/CLeAR_2023).

# B  Single Task Train

## B.1  Comparison with the previous paper

To assess the optimal performance that the model can ideally achieve, we completed Table 1 based on the results from the previous paper [6]. Each value in the table was found through extensive hyperparameter searches and 10 different random seeds and training steps up to 1,000,000. Therefore, it can be considered as the ceiling of the algorithmic reasoning performance for abstract logical tasks relying on the inductive bias of the model itself. When compared to Table 1 in the main article, which includes the mapping strategy and training with much fewer seeds and steps, our experimental results did not significantly differ and even exceeded the ceiling in some cases. However, Tape-RNN showed lower overall performance in our results, which is attributed to its sensitivity to initial seeds and relatively unstable training.

## B.2  In-distribution (ID) accuracy

Table 2 shows model accuracy on ID test data, which has input lengths of 1-40. The values in this table demonstrate higher accuracy compared to the accuracy on OOD test data, which ranges from 41 to 100, shown in the main Table 1. If a model demonstrates high accuracy on an in-distribution test set but low accuracy on an out-of-distribution test set, it indicates that the model has learned the input distribution itself rather than the generalization rule. Furthermore, it suggests that the model has already learned sufficiently from the training dataset, achieving very low training loss. Therefore, it can be inferred that it is not insufficient training but rather the inherent inductive bias of the model itself that acts as a hard barrier preventing the learning of the generalization rule in single-task learning.

Table 1: The generalization accuracy of tasks obtained by training on raw input without mapping. For accuracy over 90%, we marked the number bold. The random accuracy is 50% except for Modular Arithmetics (R), Cyclic Navigation (R), and Bucket Sort (CS). Tasks that require additional positional mapping are tagged with [†].

| Level | Task | RNN | Stack-RNN | Tape-RNN | LSTM |
|---|---|---|---|---|---|
| Regular | Even Pairs | **100.0** | **100.0** | **100.0** | **100.0** |
| | Modular Arithmetic | **100.0** | **100.0** | **100.0** | **100.0** |
| | Parity Check | **100.0** | **100.0** | **100.0** | **100.0** |
| | Cycle Navigation | **100.0** | **100.0** | **100.0** | **100.0** |
| Context Free | Compare Occurrence | **98.7** | **99.4** | **100.0** | **93.5** |
| | Stack Manipulation[†] | 56.0 | **100.0** | **100.0** | 59.1 |
| | Reverse String | 62.0 | **100.0** | **100.0** | 60.9 |
| | Divide by 2[†] | 54.0 | 81.0 | 64.0 | 51.0 |
| Context Sensitive | Duplicate String | 50.3 | 52.8 | **100.0** | 57.6 |
| | Interlocked Pairing[†] | 52.0 | **94.0** | **99.0** | **99.0** |
| | Odds First | 51.0 | 51.9 | **100.0** | 55.6 |
| | Binary Addition[†] | 50.3 | 52.7 | **100.0** | 55.5 |
| | Binary Multiplication[†] | 50.0 | 52.7 | 58.5 | 53.1 |
| | Compute Sqrt | 54.3 | 56.5 | 57.8 | 57.5 |
| | Bucket Sort | 27.9 | 78.1 | 70.7 | **99.3** |

Table 2: Baseline in-distribution accuracy (%) of overall tasks. Each model is trained for 50,000 epochs with one fixed seed. The accuracy value was reported with positional mapping. For accuracy over 90%, we marked the number bold. The random accuracy is 50% except for Modular Arithmetics (R), Cyclic Navigation (R), and Bucket Sort (CS). Tasks that require additional positional mapping are tagged with [†].

| Level | Task | RNN | Stack-RNN | Tape-RNN | LSTM |
|---|---|---|---|---|---|
| Regular | Even Pairs | **100.0** | **100.0** | **100.0** | **100.0** |
| | Modular Arithmetic | **100.0** | **99.51** | **99.99** | **100.0** |
| | Parity Check | **100.0** | **100.0** | **100.0** | **100.0** |
| | Cycle Navigation | **100.0** | **100.0** | **100.0** | **98.80** |
| Context Free | Compare Occurrence | **100.0** | **100.0** | **99.99** | **100.0** |
| | Stack Manipulation[†] | **93.87** | **99.95** | **100.0** | **99.98** |
| | Reverse String | **99.94** | **99.98** | **100.0** | **99.93** |
| | Divide by 2[†] | **99.46** | **100.0** | **99.56** | **100.0** |
| Context Sensitive | Duplicate String | 72.71 | 89.43 | **90.75** | **99.97** |
| | Interlocked Pairing[†] | **97.92** | **100.0** | **99.42** | **100.0** |
| | Odds First | 83.11 | **92.30** | **97.70** | **99.97** |
| | Binary Addition[†] | 71.52 | 81.92 | 84.13 | **98.62** |
| | Binary Multiplication[†] | 68.72 | 76.47 | 79.07 | 88.74 |
| | Compute Sqrt | 84.59 | 89.91 | **90.27** | **92.39** |
| | Bucket Sort | **99.83** | **100.0** | **94.40** | **100.0** |

## C Details in Mapping Strategy

The input sequence of an algorithmic reasoning task consists of alphabets from the corresponding formal language. Similar to how we can represent all numbers with 10 digits and create infinite sentences with 26 alphabets, the number of alphabets in a formal language is incomparably small compared to high-dimensional inputs like images. Furthermore, the mapping space, which also has discrete values of 0 and 1 for each dimension, has a limited capacity. This allows the mapping to be represented as a look-up table, thereby saving computation.

## C.1 Training mapping encoder

**Neural Net Based.** We constructed a network described in Fig 3 in the main article using a neural network to learn a mapping that satisfies the three properties in the Figure. The encoder and decoder are composed of 2-layer MLPs with ReLU activation followed by Sigmoid and the binarize function $q(\cdot)$. The entire mapping loss $L_{mapping}$ is composed of first $L_{reconstruction}$, which minimizes the difference between the input sequence and the reconstructed sequence from the Encoder-Decoder, second $L_{mean}$, which ensures that the mapped sequence from the encoder evenly occupies the entire mapping space by setting the mean of each column to 0.5, and third $L_{decorrelation}$, which sets the non-diagonal terms of the covariance matrix of mapped sequence to 0.

$$Binarize\ function\ q(x) = \begin{cases} x + stopgrad(1 - x), & \text{if } x \geq 0.5 \\ x - stopgrad(x), & \text{otherwise} \end{cases}$$

$$L_{mapping} = \lambda \cdot L_{reconstruction} + L_{mean} + L_{decorrelation}$$

$\lambda$ is set between 10-100, and a SGD optimizer with a learning rate of $1.0 \times 10^{-4}$ was used. The training was conducted with 1000 random seeds as the training is highly sensitive to initial weights. The encoder that minimizes the overall loss and achieves zero $L_{reconstruction}$ was selected. This method was applied to the `Parity Check` and `Cycle Navigation` tasks, where input sequences maintain a consistent distribution across all positions. As a result, it yielded an equivalent mapping to rule-based methods.

**Rule Based.** We utilized a rule-based approach to create mappings for all tasks which consisted of discrete input and mapping spaces. For tasks where the time steps of the input sequence are independent, we mapped each task's alphabet to be evenly distributed on the mapping spaces. When it was impossible, we made the distribution as close as possible to a uniform distribution, and also allowed the probability of 1 in a higher index column to be greater than the lower index column. For tasks where there are dependencies between each time step of the input sequence, we incorporated different methods for each task to eliminate the dependencies and enabled identical mapping across the entire space. Specific adaptation of the mappings for each task is discussed in the following section.

## C.2 Task-specific mapping methodology

The mapping strategy is represented as follows. "$0 \rightarrow (1, 0, 0), (1, 0, 1)$" On the left side are the alphabets of the input sequence, and on the right side of the arrow are the corresponding 3-dimensional samples in the mapping space. If it is a one-to-many mapping, there will be multiple corresponding samples.

`Even Pairs` (R)   $0110101 \cdots \rightarrow 0$ or 1
For consecutive pairs of numbers in the binary input sequence of 0 and 1, if the counts of the combinations "0" and "1" are equal, the label is assigned as 0. If they are different, the label is assigned as 1.
$0 \rightarrow (0, 0, 0), (0, 0, 1), (0, 1, 0), (0, 1, 1)$
$1 \rightarrow (1, 0, 0), (1, 0, 1), (1, 1, 0), (1, 1, 1)$

`Modular Arithmetic` (R)   $052746 \cdots \rightarrow N \in \{0, 1, 2 \cdots M - 1\}$
For Modular M, odd-numbered positions represent numbers from 0 to M-1, and even-numbered positions represent the operators +(M), −(M+1), ×(M+2). If the length of the input is even, the last operator is ignored. The default value for M is 5.
@Position Odd
$0 \rightarrow (0, 0, 0), (0, 0, 1)$
$1 \rightarrow (0, 1, 0), (0, 1, 1)$
$2 \rightarrow (1, 0, 0), (1, 0, 1)$
$3 \rightarrow (1, 1, 0)$
$4 \rightarrow (1, 1, 1)$
@Position Even
$5 \rightarrow (0, 0, 0), (0, 0, 1), (0, 1, 0)$

$6 \rightarrow (0, 1, 1), (1, 0, 0), (1, 0, 1)$
$7 \rightarrow (1, 1, 0), (1, 1, 1)$

## Parity Check (R)   $0110101\cdots \rightarrow 0$ or 1
If the sum of the entire sequence is even, it is labeled as 0. If it is odd, it is labeled as 1.
$0 \rightarrow (0, 0, 0), (0, 0, 1), (0, 1, 0), (0, 1, 1)$
$1 \rightarrow (1, 0, 0), (1, 0, 1), (1, 1, 0), (1, 1, 1)$

## Cycle Navigation (R)   $01020102102\cdots \rightarrow N \in \{0, 1, 2 \cdots C-1\}$
There are C points evenly spaced on a circle, numbered from 0 to C-1. Starting from 0, the label is determined by the final position obtained by repeatedly moving one step to the left, staying in place, or moving one step to the right. Left movement is represented as 0, staying in place is represented as 1, and right movement is represented as 2.
$0 \rightarrow (0, 0, 0), (0, 0, 1), (0, 1, 0)$
$1 \rightarrow (0, 1, 1), (1, 0, 0), (1, 0, 1)$
$2 \rightarrow (1, 1, 0), (1, 1, 1)$

## Compare Occurrence (CF)   $271044120\cdots \rightarrow N \in \{0, 1, 2 \cdots D\}$
For a sequence composed of digits 0, 1, $\cdots$ D, the label is determined by the number that contains the highest frequency among them. Default D=1.
$0 \rightarrow (0, 0, 0), (0, 0, 1), (0, 1, 0), (0, 1, 1)$
$1 \rightarrow (1, 0, 0), (1, 0, 1), (1, 1, 0), (1, 1, 1)$

## Stack Manipulation (CF)   $0111\cdots 243224\cdots \rightarrow 011\cdots$
A stack consisting of 0s and 1s is given, along with instructions to manipulate the stack. The output is the stack that is formed by sequentially applying the manipulation instructions to the given stack. The instructions consist of pop (2), push 0 (3), and push 1 (4). To determine the moments when the stack is modified by the instructions, additional positional mapping is required.
$0 \rightarrow (0, 0, 0), (0, 0, 1), (0, 1, 0), (0, 1, 1)$
$1 \rightarrow (1, 0, 0), (1, 0, 1), (1, 1, 0), (1, 1, 1)$
$2 \rightarrow (0, 0, 0), (0, 0, 1)$
$3 \rightarrow (0, 1, 0), (0, 1, 1), (1, 0, 0)$
$4 \rightarrow (1, 0, 1), (1, 1, 0), (1, 1, 1)$
Position of first instruction $\rightarrow 1$ in positional mapping

## Reverse String (CF)   $4521\cdots 201 \rightarrow 102\cdots 1254$
For a sequence composed of digits 0, 1, $\cdots$ D, the output is the reverse version of a string. Default D=1.
$0 \rightarrow (0, 0, 0), (0, 0, 1), (0, 1, 0), (0, 1, 1)$
$1 \rightarrow (1, 0, 0), (1, 0, 1), (1, 1, 0), (1, 1, 1)$

## Divide by 2 (CF)   $00\cdots 0100\cdots 00 \rightarrow 00\cdots 0100\cdots 00$
The input is represented in the form of $0 \times (n-1)\ 1\ 0 \times m$. In this case, the model needs to learn how to divide n by 2 and generate an output in the form of $0\times \text{ceil}(n/2)\ 1\ 0\times(\text{floor}(n/2) + m)$. In this task, the important aspect is the position where 1 exists, so the positional mapping column will have the same form as the input.
$0 \rightarrow (0, 0, 0), (0, 0, 1), (0, 1, 0), (0, 1, 1), (1, 0, 0), (1, 0, 1), (1, 1, 0), (1, 1, 1)$
$1 \rightarrow (0, 0, 0), (0, 0, 1), (0, 1, 0), (0, 1, 1), (1, 0, 0), (1, 0, 1), (1, 1, 0), (1, 1, 1)$
Position of $1 \rightarrow 1$ in positional mapping

## Duplicate String (CS)   $4521\cdots 201 \rightarrow 4521\cdots 2014521\cdots 201$
For a sequence composed of digits 0, 1, $\cdots$ D, the output is the duplicated version of a string. Default D=1.
$0 \rightarrow (0, 0, 0), (0, 0, 1), (0, 1, 0), (0, 1, 1)$
$1 \rightarrow (1, 0, 0), (1, 0, 1), (1, 1, 0), (1, 1, 1)$

## Interlocked Pairing (CS)   $0011111 \rightarrow 0011111\ 1100000$
An input of a binary sequence is given, where there are n(>0) occurrences of A ($\in \{0, 1\}$)followed by m occurrences of 1-A. In this case, the output is a sequence that follows the input sequence, having n

occurrences of 1-A and m occurrences of A. The important aspect in this task is the position of the first occurrence of 1-A in the input sequence, which determines the positional mapping column.

@Position 0

$0 \rightarrow (0,0,0), (0,0,1), (0,1,0), (0,1,1)$

$1 \rightarrow (1,0,0), (1,0,1), (1,1,0), (1,1,1)$

@Position from 1 to $l$ (length)

$0 \rightarrow (0,0,0), (0,0,1), (0,1,0), (0,1,1), (1,0,0), (1,0,1), (1,1,0), (1,1,1)$

$1 \rightarrow (0,0,0), (0,0,1), (0,1,0), (0,1,1), (1,0,0), (1,0,1), (1,1,0), (1,1,1)$

Position of first 1-A $\rightarrow$ 1 in positional mapping

**Odds First** (CS)    20125321 $\rightarrow$ 21520231

For a sequence composed of digits 0, 1, $\cdots$ D, the output is re-ordered sequence with digits in odd index come first, followed by digits in even index. Default D=1.

$0 \rightarrow (0,0,0), (0,0,1), (0,1,0), (0,1,1)$

$1 \rightarrow (1,0,0), (1,0,1), (1,1,0), (1,1,1)$

**Binary Addition** (CS)    01101011210101101 $\rightarrow$ 100011000

Two binary sequences, distinguished by a single 2, are considered as binary numbers, and their sum is outputted as a binary sequence. The position of the 2, which distinguishes the two numbers, is used as the positional mapping.

$0 \rightarrow (0,0,0), (0,0,1), (0,1,0), (0,1,1)$

$1 \rightarrow (1,0,0), (1,0,1), (1,1,0), (1,1,1)$

$2 \rightarrow (0,0,0), (0,0,1), (0,1,0), (0,1,1), (1,0,0), (1,0,1), (1,1,0), (1,1,1)$

Position of 2 $\rightarrow$ 1 in positional mapping

**Binary Multiplication** (CS)    01101011210101101 $\rightarrow$ 100100001001111

Two binary sequences, distinguished by a single 2, are considered as binary numbers, and their multiple is outputted as a binary sequence. The position of the 2, which distinguishes the two numbers, is used as the positional mapping.

$0 \rightarrow (0,0,0), (0,0,1), (0,1,0), (0,1,1)$

$1 \rightarrow (1,0,0), (1,0,1), (1,1,0), (1,1,1)$

$2 \rightarrow (0,0,0), (0,0,1), (0,1,0), (0,1,1), (1,0,0), (1,0,1), (1,1,0), (1,1,1)$

Position of 2 $\rightarrow$ 1 in positional mapping

**Compute Sqrt** (CS)    101101011 $\rightarrow$ 10011

The binary string of 0s and 1s received as input is interpreted as a binary number. The output is the binary representation of the square root (floor) of that number.

$0 \rightarrow (0,0,0), (0,0,1), (0,1,0), (0,1,1)$

$1 \rightarrow (1,0,0), (1,0,1), (1,1,0), (1,1,1)$

**Bucket Sort** (CS)    2011032022 $\rightarrow$ 0001122223

An input sequence consisting of digits from 0 to N is received, and a sorted sequence is returned. Default N=3.

$0 \rightarrow (0,0,0), (0,0,1)$

$1 \rightarrow (0,1,0), (0,1,1)$

$2 \rightarrow (1,0,0), (1,0,1)$

$3 \rightarrow (1,1,0), (1,1,1)$

## D   Comparison with Conventional CL Algorithms

In this section, we compare the performance of the conventional CL methodologies on the newly proposed CL-AR scenario. The latest CL methodologies demonstrate excellent performance in image data using techniques such as representation learning [4, 12, 11] and data augmentation [4, 3]. However, AR exhibits different characteristics from conventional image data, where applying techniques like mix-up or label smoothing can decrease accuracy. Additionally, there are challenges in applying data augmentation which is compatible with images but not with AR tasks. Therefore, we

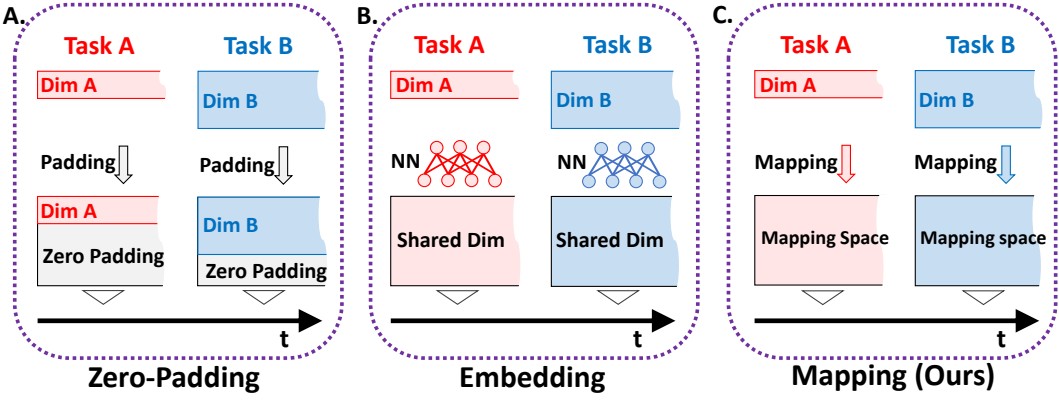

Figure 1: To enable a single model to learn tasks of various dimension sizes sequentially, the following methods were used to match the dimension of the input sequence. First, (A) zero-padding the inputs to become the same size. We set the shared input dimensions of sequential tasks to be the size of the largest dimension among all tasks. Second, (B) we used a neural network to embed inputs into the same dimension. A one-layer MLP with an activation function was used for each task. Lastly, (C) we applied a new mapping method proposed in this paper.

conduct comparative experiments on three fundamental and AR-compatible approaches: EWC [8], LwF [9], and ER [5].

### D.1    Input dimension matching

Conventional continual learning algorithms utilize models that accept fixed-size inputs. Therefore, to incorporate CL-AR tasks with dynamic input sizes, the dimensions of the inputs need to be aligned. For a more strict comparison, we matched the dimensions of the inputs using three different methods.

**Zero-padding.** The first method is the simplest, which involves matching the dimensions by zero-padding. For example, if there are tasks with dimensions of 2 and 9, we added an additional 7-dimensional zero-padding to the 2-dimensional input.

**Embedding.** The second method involves mapping the input to an intermediate layer of fixed size using a task-wise embedding network. We used a 1-layer MLP followed by ReLU activation and utilized an intermediate layer of 10 dimensions, which is much larger than our mapping dimension. This task-wise embedding is trained end-to-end.

**Mapping (Ours).** Lastly, we aligned the inputs using our newly proposed mapping methodology. This incorporates the existing CL methodologies into our novel mapping strategy.

### D.2    Experiments on conventional CL algorithms

**EWC**. Table 3 shows that default EWC ($\lambda = 1$) fails to prevent forgetting even in a very simple AR-CL scenario consisting of only four regular tasks. We thoroughly searched for the optimal coefficient of EWC's regularization term and found that even the best-performing $\lambda$, such as $\lambda = 10^{14}$ and $\lambda = 10^{16}$, exhibited very poor performance. The reason for needing such large values of $\lambda$ to compensate very small Fisher diagonal matrix is likely due to the nature of AR tasks, where models have nearly zero loss on ID datasets, resulting in a very flat loss surface and minimal Fisher diagonal matrix.

**LwF**. Table 4 shows the performance comparison between the LwF algorithm and our CLeAR method in a simple scenario of continual learning with four regular tasks. LwF showed very poor performance and prominent catastrophic forgetting in both the zero-padding method, and the embedding method, which makes input sequence distributions differ among tasks. However, our proposed mapping methodology allowed LwF to demonstrate a similar high performance to CLeAR. This is a reasonable result, considering that parts of our CLeAR algorithm resemble LwF. Nevertheless, CLeAR outperforms LwF with mapping as the tasks become more challenging, as can be seen in Table 5. When experiments were conducted on 10 tasks with varying complexities of the

Table 3: Comparison of performance with **EWC** on two CL scenarios (CN-MA-PC-EP and EP-PC-MA-CN ), averaged on 3 repeats. We reported the best performance of EWC by searching regularization coefficients $\lambda$ from 1 to $10^{21}$ and reported the best $\lambda$. The final CL accuracy was indicated in blue if its value was lower than our CLeAR method, and in red if it was higher.

| Method | Model | CN-MA-PC-EP | | | EP-PC-MA-CN | | |
| | | CL Initial | CL Final | BWT | CL Initial | CL Final | BWT |
|---|---|---|---|---|---|---|---|
| CLeAR(Ours) | RNN | 96.45 | 96.17 | -0.37 | 99.68 | 99.80 | 0.16 |
| | Stack-RNN | 94.89 | 87.09 | -10.40 | 98.19 | 99.28 | 1.45 |
| | Tape-RNN | 76.31 | 76.70 | 0.53 | 58.16 | 74.45 | 21.72 |
| | LSTM | 98.21 | 96.49 | -2.29 | 99.99 | 99.96 | -0.05 |
| EWC Default($\lambda = 1$) A. Zero-padding | RNN | 99.99 | 47.51 | —69.97 | 99.99 | 54.93 | -60.07 |
| | Stack-RNN | 99.94 | 47.57 | -69.83 | 99.99 | 54.90 | -60.12 |
| | Tape-RNN | 91.04 | 44.85 | -61.58 | 47.54 | 37.51 | -13.37 |
| | LSTM | 99.58 | 50.80 | -65.04 | 99.52 | 54.72 | -59.73 |
| EWC Default($\lambda = 1$) B. Embedding | RNN | 99.93 | 47.46 | -69.95 | 99.93 | 55.17 | -59.69 |
| | Stack-RNN | 99.86 | 47.42 | -69.93 | 99.92 | 54.72 | -60.27 |
| | Tape-RNN | 98.27 | 45.97 | -69.73 | 67.47 | 45.68 | -29.05 |
| | LSTM | 99.44 | 54.32 | -60.16 | 100.0 | 54.98 | -60.03 |
| EWC Default($\lambda = 1$) C. Mapping (Ours) | RNN | 99.64 | 47.52 | -69.49 | 98.02 | 53.50 | -59.35 |
| | Stack-RNN | 99.55 | 47.48 | -69.44 | 97.97 | 53.12 | -59.81 |
| | Tape-RNN | 87.53 | 45.15 | -56.49 | 43.82 | 37.20 | -8.82 |
| | LSTM | 99.14 | 49.63 | -66.01 | 96.83 | 51.91 | -59.90 |
| EWC ($\lambda = 10^{14}$) A. Zero-padding | RNN | 74.80 | 50.55 | -32.34 | 79.91 | 58.19 | -28.95 |
| | Stack-RNN | 74.96 | 60.32 | -19.51 | 79.54 | 57.78 | -29.02 |
| | Tape-RNN | 70.90 | 53.10 | -23.73 | 40.59 | 37.28 | -4.41 |
| | LSTM | 54.41 | 36.39 | -24.02 | 95.25 | 54.37 | -54.51 |
| EWC ($\lambda = 10^{14}$) B. Embedding | RNN | 78.85 | 58.69 | -26.87 | 80.77 | 52.38 | -37.85 |
| | Stack-RNN | 68.20 | 52.42 | -21.03 | 86.29 | 58.95 | -36.46 |
| | Tape-RNN | 78.79 | 56.52 | -29.69 | 58.54 | 48.55 | -13.33 |
| | LSTM | 54.52 | 44.41 | -13.48 | 85.28 | 60.31 | -33.30 |
| EWC ($\lambda = 10^{14}$) C. Mapping (Ours) | RNN | 54.91 | 35.04 | -26.48 | 80.77 | 52.38 | -37.85 |
| | Stack-RNN | 54.74 | 35.04 | -26.27 | 86.29 | 58.95 | -36.46 |
| | Tape-RNN | 49.32 | 34.99 | -19.10 | 58.54 | 48.55 | -13.33 |
| | LSTM | 67.37 | 44.74 | -30.17 | 85.28 | 60.31 | -33.30 |
| EWC ($\lambda = 10^{16}$) A. Zero-padding | RNN | 68.18 | 48.35 | -26.43 | 72.48 | 48.96 | -31.36 |
| | Stack-RNN | 74.84 | 57.08 | -23.67 | 75.88 | 51.09 | -33.05 |
| | Tape-RNN | 73.62 | 52.21 | -28.55 | 43.66 | 39.57 | -5.46 |
| | LSTM | 54.46 | 34.73 | -26.30 | 69.76 | 52.50 | -23.01 |
| EWC ($\lambda = 10^{16}$) B. Embedding | RNN | 73.40 | 52.63 | -27.70 | 79.78 | 54.72 | -33.42 |
| | Stack-RNN | 68.10 | 51.79 | -21.74 | 79.52 | 54.52 | -33.33 |
| | Tape-RNN | 67.51 | 45.31 | -29.60 | 67.11 | 49.55 | -23.41 |
| | LSTM | 54.29 | 34.73 | -26.08 | 73.37 | 48.39 | -33.30 |
| EWC ($\lambda = 10^{16}$) C. Mapping (Ours) | RNN | 54.94 | 35.04 | -26.53 | 60.28 | 35.26 | -33.35 |
| | Stack-RNN | 54.75 | 35.04 | -26.29 | 66.00 | 41.69 | -33.21 |
| | Tape-RNN | 49.36 | 35.04 | -19.09 | 40.91 | 35.31 | -7.47 |
| | LSTM | 60.80 | 48.33 | -16.63 | 78.93 | 53.92 | -33.34 |

Chomsky hierarchy, CLeAR exhibited outstanding performance in RNN and also showed higher performance and better BWT than LwF in LSTM.

**ER**. Table 6 shows the performance comparison between the ER algorithm and our CLeAR method on four regular tasks (left column) and dimension changing `Modular Arithmetic` tasks (right column). Due to the variety of input dimensions and lengths, it was challenging to train different tasks within a single batch. Therefore, we conducted training by alternating between the current task and the previous tasks in a 1:1 ratio. To ensure a fair comparison, we doubled the total number of training steps so that the total number of training steps for the current task remained the same. We conducted the experiments with different memory buffer sizes for storing input/output pairs: 300,

Table 4: Comparison of performance with **LwF**. We reported the results of a CL scenario consisting of the simplest four regular tasks (CN-MA-PC-EP and EP-PC-MA-CN) with 3 repeats. The final CL accuracy was indicated in blue if its value was lower than our CLeAR method, and in red if it was higher.

| Method | Model | CN-MA-PC-EP | | | EP-PC-MA-CN | | |
| --- | --- | --- | --- | --- | --- | --- | --- |
| | | CL Initial | CL Final | BWT | CL Initial | CL Final | BWT |
| CLeAR(Ours) | RNN | 96.45 | 96.17 | -0.37 | 99.68 | 99.80 | 0.16 |
| | Stack-RNN | 94.89 | 87.09 | -10.40 | 98.19 | 99.28 | 1.45 |
| | Tape-RNN | 76.31 | 76.70 | 0.53 | 58.16 | 74.45 | 21.72 |
| | LSTM | 98.21 | 96.49 | -2.29 | 99.99 | 99.96 | -0.05 |
| LwF A. Zero-padding | RNN | 99.83 | 59.95 | -53.17 | 99.86 | 55.07 | -59.71 |
| | Stack-RNN | 99.81 | 59.97 | -53.12 | 99.21 | 66.46 | -43.66 |
| | Tape-RNN | 99.20 | 59.86 | -52.45 | 83.37 | 61.69 | -28.90 |
| | LSTM | 98.51 | 58.92 | -52.78 | 99.95 | 66.63 | -44.43 |
| LwF B. Embedding | RNN | 99.84 | 47.34 | -70.00 | 100.00 | 55.02 | -59.97 |
| | Stack-RNN | 100.0 | 47.36 | -70.19 | 99.94 | 54.68 | -60.35 |
| | Tape-RNN | 98.72 | 47.47 | -68.33 | 89.42 | 54.87 | -46.07 |
| | LSTM | 99.39 | 47.56 | -69.10 | 98.60 | 53.64 | -59.95 |
| LwF C. Mapping (Ours) | RNN | 99.74 | 93.80 | -7.92 | 99.66 | 99.00 | -0.88 |
| | Stack-RNN | 97.23 | 87.70 | -12.71 | 99.79 | 98.58 | -1.62 |
| | Tape-RNN | 89.78 | 82.70 | -9.44 | 73.65 | 96.48 | 30.44 |
| | LSTM | 98.74 | 97.95 | -1.05 | 100.00 | 99.98 | -0.02 |

1000, and 3000. In a CL with four tasks, a memory buffer size of 3000 guaranteed that each task had at least 1000 samples during the whole training process, which is a very large number compared to current memory-based CL algorithms. Despite using such a large memory buffer, the ER showed lower performance than CLeAR in both scenarios.

# E Broader Impact

**Introduction of CL-AR.** For the first time, our research introduces continual learning for abstract logical concepts, which mimics the process of humans acquiring higher-order learning abilities. This moves away from the traditional CL approach centered around images and considers a new direction that CL should ultimately strive for. Algorithmic reasoning tasks are fundamentally different from image data in their abstract and logical nature. The discontinuity of input data, the necessity for generalization regarding out-of-distribution samples, and the inability to use data augmentation or mix-up techniques present the need for new CL algorithms that differ from existing methodologies. We hope that future research will further explore methodologies that effectively leverage these unique characteristics of AR.

**Tabular Data & Privacy.** Our methodology can be extended to continual learning for tabular data. One of the challenges in continual learning for table data is that the number of input columns and target columns changes for each task [10]. Our mapping strategy can be applied to handle these changing columns. Furthermore, since our mapping transforms the data into a uniform distribution in the mapping space, it can be utilized as a method to de-identify sensitive data such as tabular medical data, making it impossible to identify individuals.

Table 5: Comparison of performance with **LwF** in complex inter-hierarchy CL-AR scenario. The scenario consists of 10 AR tasks (same as main table 4) with diverse Chomsky hierarchies. Average generalization accuracy (%) is obtained for each group of tasks, repeated with 3 seeds. The final CL accuracy was indicated in blue if its value was lower than our CLeAR method, and in red if it was higher.

| Method | Model | Hierarchy | Average Accuracy (%) | | | |
| | | | Single | CL Initial | CL Final | BWT |
|---|---|---|---|---|---|---|
| CleAR (Ours) | RNN | Regular | 100.0 | 93.31 | 75.04 | -27.40 |
| | | Context Free | 77.10 | 61.59 | 60.78 | -0.81 |
| | | Context Sensitive | 57.68 | 47.13 | 46.99 | -0.19 |
| | LSTM | Regular | 97.37 | 100.0 | 93.46 | -9.06 |
| | | Context Free | 87.10 | 80.94 | 79.11 | -1.83 |
| | | Context Sensitive | 86.08 | 79.60 | 77.67 | -2.57 |
| LwF A. Zero-padding | RNN | Regular | - | 99.92 | 39.62 | -60.30 |
| | | Context Free | - | 76.49 | 49.49 | -27.00 |
| | | Context Sensitive | - | 49.01 | 42.63 | -8.51 |
| | LSTM | Regular | - | 100.0 | 39.82 | -60.18 |
| | | Context Free | - | 84.95 | 53.15 | -31.80 |
| | | Context Sensitive | - | 84.18 | 52.76 | -41.88 |
| LwF B. Embedding | RNN | Regular | - | 98.43 | 40.03 | -58.41 |
| | | Context Free | - | 71.69 | 50.36 | -21.33 |
| | | Context Sensitive | - | 46.27 | 36.85 | -12.56 |
| | LSTM | Regular | - | 98.35 | 40.10 | -58.25 |
| | | Context Free | - | 71.48 | 50.27 | -21.21 |
| | | Context Sensitive | - | 68.80 | 38.70 | -40.13 |
| LwF C. Mapping (Ours) | RNN | Regular | - | 100.0 | 40.13 | -59.87 |
| | | Context Free | - | 48.86 | 48.86 | 0.00 |
| | | Context Sensitive | - | 43.76 | 43.76 | 0.00 |
| | LSTM | Regular | - | 100.0 | 99.80 | -0.20 |
| | | Context Free | - | 84.37 | 75.51 | -8.85 |
| | | Context Sensitive | - | 84.57 | 81.06 | -4.68 |

Table 6: Comparison of performance with **ER**. We reported the results on two scenarios, each with 3 repeats. First, four regular tasks (CN-MA-PC-EP) and Second, four `Modular Arithmetics` with varying modulus from 2 to 5. Both scenario has varying size input dimension. We set the memory buffer size to 100, 300, and 3000. The final CL accuracy was indicated in blue if its value was lower than our CLeAR method, and in red if it was higher.

| Method | Model | **R**: CN-MA-PC-EP | | | **R**: MA modular 2-3-4-5 | | |
| | | CL Initial | CL Final | BWT | CL Initial | CL Final | BWT |
|---|---|---|---|---|---|---|---|
| CLeAR(Ours) | RNN | 96.45 | 96.17 | -0.37 | 98.87 | 95.88 | -2.96 |
| | Stack-RNN | 94.89 | 87.09 | -10.40 | 99.56 | 98.59 | -0.89 |
| | Tape-RNN | 76.31 | 76.70 | 0.53 | 87.13 | 83.60 | -0.55 |
| | LSTM | 98.21 | 96.49 | -2.29 | 99.98 | 99.43 | -0.19 |
| 300 memory buffer A. Zero-padding | RNN | 99.81 | 49.67 | -66.85 | 99.45 | 51.65 | -63.74 |
| | Stack-RNN | 99.89 | 51.26 | -64.84 | 99.18 | 51.62 | -63.42 |
| | Tape-RNN | 70.98 | 40.47 | -40.68 | 40.73 | 32.26 | -11.29 |
| | LSTM | 99.59 | 55.75 | -58.46 | 99.95 | 59.21 | -54.31 |
| 300 memory buffer B. Embedding | RNN | 99.83 | 47.58 | -69.66 | 95.63 | 47.85 | -63.71 |
| | Stack-RNN | 99.87 | 57.23 | -56.85 | 92.46 | 44.69 | -63.69 |
| | Tape-RNN | 81.51 | 42.32 | -52.26 | 45.40 | 32.48 | -17.23 |
| | LSTM | 99.44 | 52.21 | -62.97 | 99.83 | 58.95 | -54.51 |
| 300 memory buffer C. Mapping (Ours) | RNN | 95.53 | 47.42 | -64.14 | 97.90 | 50.15 | -63.67 |
| | Stack-RNN | 99.72 | 47.54 | -69.57 | 99.82 | 52.15 | -63.56 |
| | Tape-RNN | 80.06 | 42.14 | -50.57 | 43.57 | 32.34 | -14.98 |
| | LSTM | 99.46 | 56.11 | -57.79 | 99.96 | 52.29 | -63.57 |
| 1000 memory buffer A. Zero-padding | RNN | 99.94 | 55.96 | -58.63 | 82.33 | 34.77 | -63.41 |
| | Stack-RNN | 99.42 | 57.76 | -55.55 | 99.21 | 60.72 | -51.33 |
| | Tape-RNN | 74.24 | 44.77 | -39.30 | 41.50 | 32.05 | -12.59 |
| | LSTM | 99.58 | 63.39 | -48.26 | 99.57 | 65.07 | -46.01 |
| 1000 memory buffer B. Embedding | RNN | 99.96 | 78.34 | -28.83 | 93.10 | 55.51 | -50.12 |
| | Stack-RNN | 99.89 | 77.26 | -30.18 | 92.02 | 55.34 | -48.91 |
| | Tape-RNN | 91.66 | 54.32 | -49.80 | 44.89 | 32.67 | -16.29 |
| | LSTM | 98.65 | 63.31 | -47.12 | 99.98 | 70.17 | -39.74 |
| 1000 memory buffer C. Mapping (Ours) | RNN | 81.69 | 51.67 | -40.02 | 98.87 | 51.35 | -63.35 |
| | Stack-RNN | 86.66 | 49.69 | -49.29 | 99.81 | 52.20 | -63.49 |
| | Tape-RNN | 69.44 | 42.83 | -35.48 | 45.27 | 32.78 | -16.65 |
| | LSTM | 99.44 | 56.35 | -57.46 | 99.95 | 52.25 | -63.60 |
| 3000 memory buffer A. Zero-padding | RNN | 99.96 | 77.90 | -29.43 | 94.53 | 63.18 | -41.80 |
| | Stack-RNN | 99.96 | 77.93 | -29.38 | 98.07 | 71.03 | -36.06 |
| | Tape-RNN | 74.90 | 45.87 | -38.70 | 47.11 | 32.65 | -19.29 |
| | LSTM | 99.57 | 75.86 | -31.61 | 99.97 | 86.01 | -18.61 |
| 3000 memory buffer B. Embedding | RNN | 99.78 | 79.90 | -26.51 | 99.77 | 87.13 | -16.84 |
| | Stack-RNN | 99.98 | 82.46 | -23.36 | 99.21 | 88.96 | -13.68 |
| | Tape-RNN | 88.14 | 59.62 | -38.04 | 43.58 | 34.51 | -12.09 |
| | LSTM | 99.45 | 73.66 | -34.38 | 99.99 | 92.92 | -9.43 |
| 3000 memory buffer C. Mapping (Ours) | RNN | 80.14 | 56.59 | -31.39 | 99.41 | 51.86 | -63.40 |
| | Stack-RNN | 90.93 | 56.37 | -46.08 | 99.67 | 52.09 | -63.44 |
| | Tape-RNN | 63.94 | 42.62 | -28.43 | 57.35 | 35.55 | -29.06 |
| | LSTM | 98.21 | 62.47 | -47.65 | 99.86 | 55.97 | -58.52 |