# OpenReview forum: "CLeAR: Continual Learning on Algorithmic Reasoning for Human-like Intelligence"
_NeurIPS.cc/2023/Conference — NeurIPS 2023 poster_

### Official Review · Reviewer_aCtv · 2023-06-29

**Soundness:** 3 good
**Presentation:** 2 fair
**Contribution:** 2 fair
**Rating:** 5
**Confidence:** 4

**Summary:**

This work tackles the problem of continual learning for rule-based tasks. The method first maps algorithmic inputs to a discrete latent space. This mapping is one-to-many, which allows the method to learn multiple features for the inputs of each task. A model then consumes results from this discrete latent space to perform the task. The authors test and verify the method works on various algorithmic tasks drawn from the Chomsky hierarchy.

**Strengths:**

- The authors define a new set of problems for continual learning
- The approach for mapping inputs to learned features makes sense
- Continual learning is becoming more prevalent as models become more expensive to train from scratch, so the work is timely

**Weaknesses:**

- The framing uses catastrophic forgetting, but whether catastrophic forgetting is a pressing issue in today’s deep models is an open question. We do not know how increasing model scale impacts catastrophic forgetting, and deep networks like GPT-3 are fine-tuned often in production. Recent empirical results on catastrophic forgetting in neural networks are not as dramatic as the authors imply, and these works should be cited. See my comments in the questions section for references.
- It’s unclear how the experiments in the paper map onto real world problems. Can the authors say more about this? When would we want to train a model sequentially over algorithmic tasks?
- It seems CLeAR gets outperformed by multitask and single task training. Is there a baseline that would demonstrate the lift of this approach?

See questions for other concerns.

**Questions:**

Major questions:
- Line 229: How do you control the size of the mapping set?
- How much of the performance is explained by learning regular languages well and nothing else? For example, in the context-free tasks, are the models just solving the subset of examples that can be modeled via a regular language?
- Is the CL setting better than the pretrain-finetune or meta-learning settings, where we might pretrain/meta-train over multiple tasks and then adapt to one task?

Minor
- Line 24: The decline is not necessarily rapid. See [this](https://arxiv.org/abs/1910.08475) [this](https://arxiv.org/abs/2102.07686) for more recent citations to catastrophic forgetting.
- Line 113: “the model performing backward transfer is still limited.” What does this mean?
- Line 126: Missing citations to [this](https://arxiv.org/abs/2106.16213) [this](https://aclanthology.org/2020.acl-main.43/) and related works on the languages that can be recognized by transformers and RNNs. These works essentially define the upper bounds to your empirical work.
- Table 3: What is the logic behind the bold numbers here?

Nits:
- Line 16: typo “during following”
- Line 83: could you spell out LwF?
- Line 365: duplicated references

**Limitations:**

I would argue that an additional limitation is the latent space design itself. More accurately, the discrete latent space comes with tradeoffs. Because the method does not define a distance metric over the discrete latent space, there is no notion of exemplars or typicality in the mapped sequences. By design, all the sequences represent features of equal importance. However, one might presume that for some tasks, certain features of task deserve a higher weighting than others. This seems easier to solve in a continuous latent space, where you can map the importance of a feature onto distance.

---

> ### Author Rebuttal · Authors · 2023-08-10
>
> *We thank you for the positive comments and suggestions. We have addressed each of your questions below.*
> ***
> ## Weaknesses
> **W1** We appreciate your insightful analysis on catastrophic forgetting, especially on recent LLMs. Following your suggestion, we have incorporated the references you mentioned into our work (Global Comment Literature Reviews 2).
> ***
> **W2** So far, CL-AR has been a challenging field where deep learning models have not yet demonstrated practical real-world performance, making it difficult to apply in everyday life.
>
> In the most ideal scenario, a robot, like a human, continuously acquires logical knowledge while integrating previous knowledge without losing it. In more practical fields, medical/industry datasets containing information that requires logical reasoning may be benefitted from CL-AR. Especially when it's not possible to define the task all at once from the beginning or access to previous data is prohibited due to privacy issues.
>
> In a broader scope, we believe that our method of continually training logics could contribute to future LLM training like Open AI is now trying to enhance its performance on reasoning tasks through process supervision.
>
> Ultimately, we expect that through the challenging CL-AR learning scenario that current neural networks struggle with, we may gain insights into how humans learn and integrate knowledge in the brain.
> ***
> **W3** To establish a baseline for assessing the performance of our model, we attempted the following three approaches.
> First, we compared the performance of our CLeAR method with the Conventional CL algorithm as a baseline. The performance of the EWC, LwF, and ER methods is indicated in Supplementary D, demonstrating that our CLeAR significantly outperforms these methods.
>
> Second, to further demonstrate the lift of our approach, we calculated the accuracy difference after the initial training of each task and after all tasks, which indicates how much information the model lost by CL. CLeAR not only retained nearly all information but also frequently exhibited backward transfer of information to previous tasks.
>
> Third, according to the paper “A continual learning survey:Defying forgetting in classification tasks(2021)” multitask training (joint training) is considered as a soft upper bound in CL scenario. And for most of the conventional cl methods, there is a considerable performance gap between multitasking training. Although our work is the first paper on CL scenarios in a completely new field, approaching joint training performance still indicates excellent continual learning is taking place. In fact, our model exhibited very high performance close to joint training and even demonstrated impressive results surpassing joint training performance on multiple tasks.
> ***
> ## Questions
> **Q1** We apologize for the insufficient information transfer to the model. We understood the “mapping set” you mentioned as the dimension of mapping space. The dimension of the mapping space is a hyperparameter determined by us, similar to the model's hidden dimension. It is advisable to set a larger value for the dimension when the number of alphabets in the input data is high. Our mapping function aligns the raw input sequence to the defined mapping space dimension. Furthermore, it is possible to increase the dimension of the mapping space during CL to readjust its size.
> ***
> **Q2** According to formal language theory, tasks from the upper Chomsky hierarchy (context-free tasks) can not be represented by grammar on the lower Chomsky hierarchy  (regular grammar) or solved by corresponding automata (Finite-state automata).
> Therefore, even if finite state automata produced correct answers in some samples, It is appropriate to attribute this to random chance. However, theoretically, a neural network is a universal approximator capable of solving all problems, akin to a Turing machine. Consequently, if a neural network generates correct answers for certain samples, further analysis is necessary to determine whether it is due to random chance or if the network has indeed learned certain aspects of the task.
> ***
> **Q3** CL, pretrain-finetune (P-F), and meta-learning (M-L) address different learning scenarios. It seems difficult to say which setting is better. In the context of AR, the goal is to excel at individual tasks rather than creating a generally superior model, which is why we found M-L incompatible. Additionally, due to the shifting data distributions for each task and the need to adapt well to previously learned tasks, we considered CL more suitable than P-F. During this rebuttal, we conducted additional experiments and discovered possibilities that pretraining on multiple tasks might not be advantageous in learning multiple AR tasks, making our CL-AR setup worthwhile. (Please refer to **General Response Experiment 2 and iPYe Q1** for detail).
> ***
> **Q4** Apologies for the insufficient description. This statement implies that even recent CL methodologies find achieving positive backward transfer quite challenging. Recent methods such as AFEC (NeurIPS 2021) achieve only slight positive backward transfer on simple image datasets.
> ***
> **Q5** Thank you for introducing important papers related to this study. As per your comment, we will make sure to cite the two mentioned papers in the related work section of the final version (**Global Comment Literature Reviews 3**).
> ***
> **Q6** For the CL Finals column, the emphasis is on results above 90%, and for BWT and FWT columns, on positive values. However, the non-bold case does not imply low CL performance. Due to the text limit, please refer to **iPYe W1** for a detailed explanation.
> ***
> ## Limitations
> **L1** We agree with your statement. Expanding this work into distance-preserving mapping in a continuous feature space would be meaningful. As part of future work, we intend to explore our continuous feature space mapping using models such as diffusion models.

---

> > ### Author Response · Authors · 2023-08-18
> > **Looking forward to your post-rebuttal feedback**
> >
> > Dear Reviewer aCtv
> >
> > Thank you again for the insightful comments and suggestions. Since the deadline for our discussion is approaching, we sincerely look forward to your follow-up response. We are happy to provide any additional clarification that you may need.
> >
> > For your convenience, we provide a summary below:
> >
> > * #### A description has been provided about how CL-AR tasks will be utilized in the real world, with connections to LLMs and human learning.
> >
> > * #### Discuss the phenomenon of CL often surpassing multitask learning In CL-AR (multitask learning is a soft upper bound in traditional image-based CL).
> >
> > * #### To systematically analyze the advantages of CL, we newly devise a novel approach to analyze what kind of hierarchical representation or compression of previous "knowledge" is happening to enable CL (attached **PDF & iPYe Q1**).
> >
> > * #### The method for adjusting the size of the mapping set, and whether context-free tasks can be explained by regular languages from the perspective of formal language theory.
> >
> > * #### References have been added to LLM's catastrophic forgetting and previous AR-related research.
> >
> > * #### Enhanced the aspects of the paper where the explanation was insufficient.
> >
> > We hope that the provided new experiments and the additional discussion have convinced you of the merits of this paper. Please do not hesitate to contact us if there are additional questions.
> >
> > Meanwhile, we thank you for your very helpful comments. It would indeed make our paper clearer and stronger.
> >
> > Thank you for your time and effort.
> >
> > Best regards, Authors

---

> ### Author Response · Authors · 2023-08-21
> **A kind reminder**
>
> ### Dear reviewer aCtv
>
> We wanted to kindly remind you that the interactive discussion phase will be ending in just a few hours. Unfortunately, we won't be able to engage in further discussions with you after the deadline. We hope that our response has addressed your concerns, and turned your assessment to a more positive side. Please let us know if there are any other things that we need to clarify.
>
> We thank you so much for your helpful and insightful suggestion.
>
> Best, Authors

---

### Official Review · Reviewer_GYfA · 2023-07-04

**Soundness:** 3 good
**Presentation:** 3 good
**Contribution:** 3 good
**Rating:** 6
**Confidence:** 4

**Summary:**

This paper introduces CLeAR, a novel algorithmic reasoning (AR) methodology for continual learning (CL) of abstract logical concepts. The main component of CLeAR is a one-to-many mapping strategy that aligns various tasks of different dimensions in a shared mapping space, allowing for the gradual learning of abstract concepts. The paper addresses the challenges specific to CL for AR tasks, such as decorrelated input data, dynamic dimensions of datasets, and the goal of generalization for out-of-distribution data. Extensive experiments are conducted, demonstrating that CLeAR achieves near-zero forgetting and improved accuracy during subsequent training, outperforming existing CL methods designed for image classification. The paper highlights the importance of studying CL in the context of abstract logical concepts and offers valuable insights into human learning mechanisms.

**Strengths:**

The paper introduces a novel methodology, CLeAR, for continual learning of abstract logical concepts. It addresses the lack of research in this area and presents a unique approach to aligning tasks of different dimensions and sharing semantics. By focusing on abstract algorithmic reasoning tasks, the paper bridges the gap between CL and real-world cognitive skills development. The proposed one-to-many mapping strategy addresses the challenges unique to abstract concept learning and offers a solution for preventing forgetting and improving accuracy during subsequent training.

**Weaknesses:**

From my point of view, I'm not sure why end-to-end algorithmic reasoning is important when this problem can easily be solved via symbolic reasoning with pre-defined rules and structures. Also, I think the paper should include some other baseline experiments, like standard CL methods, for comparison. Foundation models can also be considered if possible. This would provide a better understanding of the performance improvement achieved by CLeAR and allow for a more comprehensive assessment of its effectiveness.

**Questions:**

- Could the authors provide further insights on why end-to-end approaches are necessary or beneficial in comparison to symbolic reasoning with pre-defined rules and structures?
- Have the authors considered including other CL methods or foundation models in the evaluation? I guess most of the tasks listed in the paper can be solved by models like gpt4. So why learning Algorithmic Reasoning important?

**Limitations:**

See weaknesses and questions.

---

> ### Author Rebuttal · Authors · 2023-08-10
>
> *We thank you for the positive comments and suggestions. We have addressed each of your questions below.*
> ***
> ## Weaknesses
> **W1** (Aligned with Question 1)  We appreciate your insightful comment. Frameworks such as **DeepProbLog (2018), HOUDINI (2018), NeuralTerpret (2017)**, and others have been developed using symbolic reasoning with predefined rules and structures to solve AR. We intend to explain the significance of our end-to-end algorithmic reasoning by focusing on the differences between our work and these frameworks.
>
> Firstly, our model does not rely on any human (prior) knowledge about the tasks that the model will learn. The model purely performs "Learning from data" without knowing what logical tasks it is dealing with. In contrast, the aforementioned frameworks require pre-configured programs that handle logic using human knowledge of each task. This contradicts the assumption of continual learning, as it is impossible to anticipate and program for all potential problems that may arise in the real world.
>
> Secondly, our paper performs logical operations using a pure neural network. While the aforementioned frameworks do utilize neural network structures internally, they mainly use them as feature extractors, such as extracting features from MNIST digits or recognizing arrow directions. Even in lifelong learning scenarios presented in those frameworks (e.g., tasks like summing two images to performing arithmetic operations on multiple images), the neural network acts as a feature extractor, while pre-defined programs perform the actual logical operations.
>
> Lastly, the mapping methodology proposed in our paper allows for continual learning of multiple tasks with different dimensions, relying solely on data without any prior knowledge about the tasks. While the aforementioned papers also introduce sequential learning scenarios, they lack scalability as they either rearrange modularized programs for each task or create predefined parameterized programs using knowledge of all the future tasks from the beginning.
>
> Our research proposes a method where the model can learn any task without the need for human intervention to change the model's structure for each task. This is similar to how a person who has learned arithmetic can smoothly integrate previous knowledge when learning linear algebra without rewiring the brain, approaching the key goal of CL, which is to learn consecutive tasks like a human.
> ***
> **W2** We completely agree with what you said. The comparison results with the three baseline methods are currently only presented in the supplementary section D. As a result, we observed that CLeAR outperforms all other CL methods in terms of final CL accuracy and knowledge backward transfer. In the final version of the paper, we will mention the comparison results of the three baseline CL methodologies, **EWC (Table 3 supple.), LwF (Table 4,5 supple.), and ER (Table 6 supple.)**, in the main paper.
> ***
> **W3** We appreciate your inspiring comment. Recently in the field of CL, there is a growing trend of using foundation models like LLM (large language model). Based on your comment, first of all, to obtain a deeper understanding of our model's functioning, we conducted a more comprehensive assessment using the new analytical methods that we propose. Through this, we have discovered that our CLeAR learns reasoning by internally simulating automata and is capable of integrating automata states for new tasks. Additionally, we have analyzed that it can exhibit higher learning capability than what is typically considered an upper bound in joint training in the context of Continual Learning (Global Comment Experiments 2). In future work, as LLM has demonstrated high performance across various domains, applying a foundation model to CL-AR tasks would be a novel research direction. In future work, we will attempt CL-AR task learning and evaluation using the LLM approach, utilizing resources like the gpt4 API or Alpaca (2023).
> ***
> ## Questions
> **Q1** Explained in W1
> ***
> **Q2-1** As mentioned in W2, we compared the performance of three baseline CL methods using three different mapping methods and provided the results in the Supplementary Material.
> ***
> **Q2-2**  In addition to its pure academic value in studying the process of logic learning in models, AR also holds significance in relation to LLM for the following reasons.
>
> According to recent research on Transformer architecture and algorithmic reasoning (ref[11,24] in main and "Simplicity Bias in Transformers and Their Ability to Learn Sparse Boolean Functions" (2022)), Transformers often struggle with generalization to out-of-distribution (OOD) data of varying lengths due to their tendency to learn shortcuts in AR tasks.
>
> For instance, regardless of the number of model heads, hidden dimensions, or training epochs, and the type of positional embedding, transformers fail to learn algorithms of very simple (for human) tasks like Parity check (PC: whether the sum of a binary sequence is even or odd). Despite showing 100% accuracy on test sets of the same length as the training data, the model's accuracy drops to a random chance of 50% for longer OOD sequences. This is attributed to the Transformer architecture's focus on memorizing input data distribution rather than learning rules, leading to shortcut learning. From this perspective, LLMs with Transformer architecture could also encounter similar challenges. In a broader scope, we think that a methodology of continually training our AR task could potentially offer new insights to enhance the model architecture of LLM like OpenAI's recent attempts of step-by-step learning through process supervision.
> Ultimately, we expect that through the challenging CL-AR learning scenario that current neural networks struggle with, we may gain insights into how humans learn and integrate knowledge in the brain.

---

> > ### Author Response · Authors · 2023-08-18
> > **Looking forward to your post-rebuttal feedback**
> >
> > Dear Reviewer GYfA
> >
> > Thank you again for the insightful comments and suggestions. Since the deadline for our discussion is approaching, we sincerely look forward to your follow-up response. We are happy to provide any additional clarification that you may need.
> >
> > For your convenience, we provide a summary below:
> >
> > * #### Introduction to the conventional neuro-symbolic framework with NNs and why these methodologies are infeasible in CL scenarios, leading to the clear necessity of using our pure NN-based CLeAR approach (with **Literature reviews 1 in global comment**).
> >
> > * #### Enhanced comparison with baseline performance of conventional CL methodologies.
> >
> > * #### Development of a novel assessment method to achieve a more comprehensive evaluation of the performance improvement achieved by CLeAR, along with in-depth analysis using this method (details in the attached **PDF & iPYe Q1**).
> >
> > * #### The weaknesses of the current transformer model in handling AR tasks and the potential for CL-AR models to be utilized in training reasoning capabilities of future LLMs.
> >
> > We hope that the provided new experiments and the additional discussion have convinced you of the merits of this paper. Please do not hesitate to contact us if there are additional questions.
> >
> > Meanwhile, we thank you for your very helpful comments. It would indeed make our paper clearer and stronger.
> >
> > Thank you for your time and effort.
> >
> > Best regards, Authors

---

> ### Author Response · Authors · 2023-08-21
> **A kind reminder**
>
> ### Dear reviewer GYfA
>
> We wanted to kindly remind you that the interactive discussion phase will be ending in just a few hours. Unfortunately, we won't be able to engage in further discussions with you after the deadline. We hope that our response has addressed your concerns, and turned your assessment to a more positive side. Please let us know if there are any other things that we need to clarify.
>
> We thank you so much for your helpful and insightful suggestion.
>
> Best, Authors

---

### Official Review · Reviewer_iPYe · 2023-07-07

**Soundness:** 4 excellent
**Presentation:** 4 excellent
**Contribution:** 4 excellent
**Rating:** 7
**Confidence:** 3

**Summary:**

The authors propose a training framework for continual learning in which task inputs are mapped to regions of or distributions over an embedding space, before the resulting embedding is passed to a single model that learns continually to solve the tasks.  They apply this continual learning framework to tasks based on learning algorithms, testing against input-output pairs from outside the training distribution. The resulting training and task framework can solve a number of algorithmic reasoning tasks, up to the limitations imposed by model architectures.

**Strengths:**

The paper is one of the first I've seen to tackle algorithmic reasoning/algorithm induction tasks along with continual learning. The core idea of mapping tasks and inputs to an embedding space from which a unified model learns continually to solve tasks (via a task-specific output head) is, to my knowledge, highly novel. The paper is duly careful about which levels of the Chomsky hierarchy it considers as algorithms for purposes of defining the algorithmic reasoning task.

**Weaknesses:**

The paper needs to more clearly present its comparisons between a baseline and a contribution.  I understand that the authors do not claim every task on which they evaluated to be "solved", especially since many of them require positional encodings and a context-free or context-sensitive memory.  However, since the message is more complicated, the authors could do a better job presenting exactly where their training framework and meta-model architecture makes a difference against, for instance, just training the baseline neural network architectures with no continual learning, or training them and then retraining with a baseline level of catastrophic forgetting.

The authors have since conducted a number of additional experiments to address this weakness.

**Questions:**

Where do the authors think some form of hierarchical representation or compression of previous "knowledge" is happening to enable continual learning?

The authors have submitted supplementary material on exactly this question.

**Limitations:**

The authors duly discuss the chief limitation of their approach being a discretization in the embedding space.

---

> ### Author Rebuttal · Authors · 2023-08-09
>
> *We thank you for the positive comments and suggestions. We have addressed each of your questions below.*
> ***
> ## Weakness
> **W1** Thank you for the insightful comments. We apologize for the insufficient explanation of the experimental results. As you suggested, we will address these points in the final version of the paper.
>
> **Explanation**
> In single task training (Table 1 main), an accuracy exceeding 90% was considered that the model learned the AR. In such cases, the model possesses accurate algorithmic information, making it easier to transfer this knowledge to subsequent tasks. However, many models and tasks exhibit low accuracy in single task training. To make CLeAR an effective CL methodology, we believe it is crucial to preserve even partial information (ACC < 90%) when the model is not able to learn the algorithm perfectly and transfer it successfully. Therefore, we conducted experiments even for challenging tasks with low single-training performance. Performance of CLeAR is evaluated through CL Initial, CL Final, and BWT values. The average of single task training results for each task and the joint training performance, which is generally considered an upper bound in typical CL scenarios, provide insights into the difficulty of tasks, rather than just CLeAR's performance.
>
> Across our experiments, CLeAR demonstrated the ability to preserve even incomplete information (CL Initial < 90%) for challenging tasks with a high Chomsky hierarchy. First, for difficult tasks, the CL Final score did not significantly decrease compared to the initial CL Initial score, and in many cases, performance even increased through BWT. Second, CLeAR outperformed the baseline of existing CL methodologies (supple. D).
>
> From these observations, we can conclude that our CLeAR methodology is valid not only for learning individual tasks but also for difficult tasks with a high Chomsky hierarchy.
> ***
> ## Questions:
> **Q1** We sincerely appreciate your crucial question. Thanks to your comment, we devise new analytical methods to address a crucial yet challenging aspect of CL: how the model preserves and integrates information from previous tasks while incorporating new knowledge.
> In addition, we found interesting results with your comment (W1). Unlike typical CL scenarios where joint training is an upper bound, our experiments often showed lower results compared to CL. Analyzing the model's internal learning process in joint training, we found that while CLeAR acts as a regularizer, integrating information between tasks, joint training in CL-AR fails to find common information.
> These new analyses advance the paper a step further and highlight the significance of CL in AR tasks, similar to human learning logic in real-world situations.
>
> **Experiments (Fig.R1 and Fig.R2 in PDF)**
> We conducted experiments with two simple regular tasks, Parity-check (PC: whether the sum of input is even or odd) and Even-pairs (EP: whether the count of two consecutive digit pairs, 01 and 10, is the same), both with binary inputs, using a fixed seed and an RNN model. In both CL and joint training, 100% IID and OOD accuracy was achieved.
>
> From the perspective of formal language theory, learning each task requires acquiring the automata structure (Fig.R1 D) internally. Fig.R1 A, B, C, E represent the model's hidden feature space colored corresponding to PC (up) and EP (down) automata states for the input sequence. Fig.R1 A shows that after learning PC, the model has fully acquired the corresponding automata (up), while for the unlearned EP, the points remain undivided (down). Fig.R1 B demonstrates that after learning EP via CLeAR, the model maintains information from PC (upper) while learning EP (lower), with minimal change in feature space. The first task acts as a regularizer, preventing inefficiency and maintaining a minimal joint automata state for both tasks.
>
> Fig.R1 C illustrates joint training, where the feature space of the model is separated for the corresponding automata state (up), leading to the use of less effective feature space. Opposed to CL, it failed to learn commonality. This characteristic is more distinct in Fig.R1 E. This is an easy-to-learn feature for the model but doesn't contribute to problem-solving, making it unnecessary to learn. Fig.R1 E shows that while joint training leads to unnecessary clusters, CL avoids learning redundant features. These observations suggest that CLeAR acts as a regularizer in CL-AR, reducing the ineffective use of feature space and integrating information between tasks by combining corresponding automata states.
>
> For more complex context-free (Stack manipulation: SM) and context-sensitive (Bucket sort: BS) tasks, Stack-RNN was used (Fig.R2). In the case of SM, for learning to occur, the model must necessarily internalize the usage of a memory stack. CL exhibited superior results compared to joint training for both SM and BS in IID and OOD test sets. To verify whether the model has indeed learned true algorithms in each case, Fig.R2 B shows the memory stack actions based on the input alphabet. In CL (left), it is evident that the model has accurately learned the stack usage. Furthermore, during the learning process of BS, it retains this knowledge and even gets closer to appropriate actions, resulting in backward transfer. Fig.R2 C displays stack changes based on the sequence in both CL and joint training, revealing more appropriate learning in CL, while what joint training learned is a shortcut for IID, not the actual algorithm.
>
> Through these experiments, CL-AR was shown to integrate task-related information by learning automata internally and positioning similar-role states closely in the feature space. Additionally, CL could be more preserving and robust against shortcut learning compared to joint training. This work showcased the strength of CL methodology in learning logics  and proposed a novel approach for analyzing internal learning processes from a CL perspective.

---

> > ### Author Response · Authors · 2023-08-18
> > **Looking forward to your post-rebuttal feedback**
> >
> > Dear Reviewer iPYe
> >
> > Thank you again for the insightful comments and suggestions. Since the deadline for our discussion is approaching, we sincerely look forward to your follow-up response. We are happy to provide any additional clarification that you may need.
> >
> > For your convenience, we provide a summary below:
> >
> > * #### Strengthened experimental explanation and showed evidence for the strong performance of CLeAR across tasks of various difficulties.
> >
> > * #### Enhanced comparison with baseline performance without using CL and existing CL methodologies.
> >
> > * #### Newly devise a novel approach based on CL-AR to analyze what kind of hierarchical representation or compression of previous "knowledge" is happening to enable CL (with **attached PDF**)
> >
> > We hope that the provided new experiments and the additional discussion have convinced you of the merits of this paper. Please do not hesitate to contact us if there are additional questions.
> >
> > Meanwhile, we thank you for your very helpful comments. It would indeed make our paper clearer and stronger.
> >
> > Thank you for your time and effort.
> >
> > Best regards, Authors

---

> ### Author Response · Authors · 2023-08-21
> **A kind reminder**
>
> ### Dear reviewer iPYe
>
> We wanted to kindly remind you that the interactive discussion phase will be ending in just a few hours. Unfortunately, we won't be able to engage in further discussions with you after the deadline. We hope that our response has addressed your concerns, and turned your assessment to a more positive side. Please let us know if there are any other things that we need to clarify.
>
> We thank you so much for your helpful and insightful suggestion.
>
> Best, Authors

---

> > ### Comment · Reviewer_iPYe · 2023-08-21
> >
> > The authors have provided a satisfactory response to my concerns, and I am intrigued to explore their additional experiments.

---

> > > ### Author Response · Authors · 2023-08-21
> > > **Thank you for your positive reply**
> > >
> > > We sincerely appreciate your reviews. We're truly pleased that our responses satisfactorily addressed your concerns.
> > > Thank you very much for raising the score and viewing our additional experiments favorably.
> > >
> > > Thanks to your insightful comments, we were able to strengthen the paper significantly.
> > >
> > > Best, Authors

---

### Official Review · Reviewer_iTug · 2023-07-09

**Soundness:** 3 good
**Presentation:** 3 good
**Contribution:** 3 good
**Rating:** 6
**Confidence:** 3

**Summary:**

In this paper, the authors introduce CLeAR, a new method of Continual Learning (CL) for Algorithmic Reasoning (AR). In doing so, several relaxations to standard CL are discussed: (1) scenarios where the same data are used for different tasks, (2) the variation of the input, being not of fixed size, and (3) analysis of performances on OOD data. The main algorithm proposed, namely CLeAR, combines a map to a shared feature space with a RCNN or a LSTM where learning happens, and uses an additional MLP to project in the output space. The CL strategy for catastrophic forgetting builds on LwF.

To evaluate the proposed model, the authors propose 15 benchmarks of increasing difficulty, based on the use of different Chomsky hierarchies. The experiments are conducted for the method and several baselines, some of which include known CL strategies. Overall the results show that CLeAR addresses the CL in AR, providing also positive Backward transfer.

**Strengths:**

The authors address for the first time the CL adaptation of AR. The setup proposed differentiates from standard CL, where input data dimensions are fixed, and evaluation is typically done in an IID test-set w.r.t. to the joint task distribution. This leads to a more general setting w.r.t. standard CL which can improve current research and lead to new methodologies.

The proposed method well adapts to variegated AR tasks, being designed to use a shared feature space where learning takes place. The experiments show that this choice has the desired effects, addressing catastrophic forgetting in several scenarios. In addition, the newly designed benchmarks (to be released on github) offer space for further research in this new field.

The presentation is overall clear, with much focus on benchmarking the method in several experiments.

**Weaknesses:**

It seems some information is missing in the current text, and the experimental evaluation can be improved. Referring to Table 3, the discussion in Section 4.2 only mentions the first 4 rows, but nothing is mentioned about the last three. How should those results be interpreted and what do they convey? It seems that the method is not solving the task, accordingly to the authors' claim that the threshold should be above 90% of the task's accuracy.

As a suggestion, the comparison with existing CL strategies is very important and should appear or at least be discussed in the main text.

All experiments are conducted in a single run, contrary to common practice, especially in CL. This may lead to a confirmation bias over the proposed method and should be clarified at least here why the choice of single runs does convey all relevant information. Based on the experiments in Appendix D, I'm not entirely sure that LwF combined with the shared feature mapping would drastically differentiate from CLeAR (Tables 4 and 5), since it is showing very similar classification scores. To this end, I sense that adopting LwF is particularly suited for the task the authors are proposing since it is likely adapting previous tasks to new observed sequences.

**Typo** In the bibliography refs [33] and [34] coincide.

**Questions:**

Can you provide more details about why LwF is performing very well in CL-AR? Is it because of the pseudo-labels in the initial training phase?

**Limitations:**

**Major** The related work connected to AR is poorly discussed and only one recent work is mentioned in the related work. This gives somehow the feeling that research in AR is niche. To this end, more works should be cited or, otherwise, the motivation for CL in AR should be strengthened.

**Minor** Details on the benchmarks are not entirely clear, making a bit fuzzy the decision behind their design. Such explanation can be improved even in the Supplementary to make understand how different tasks combine in increasing level of difficulty

---

> ### Author Rebuttal · Authors · 2023-08-09
>
> *We thank you for the insightful comments and suggestions. We have addressed each of your questions below.*
> ***
> ## Weaknesses
>
> **W1** Thank you for your sharp comments. And we apologize for the insufficient explanations in certain parts. As you pointed out, we will make sure to incorporate the mentioned content into the final version of the paper.
>
> The accuracy exceeding 90% in the single training (Table 1) indicates that the model has learned the algorithm perfectly. In this case, the model possesses accurate information about the algorithm, making it easier to transfer this knowledge to the next task. However, to create an excellent CL methodology, we believe it is important to preserve and transfer even partial information (CL Initial under 90%) about the task, even if it is not complete. Therefore, rows 5, 6, and 7 in Table 3 serve as positive indicators that the model can preserve even incomplete information for challenging tasks with a high Chomsky hierarchy. Firstly, the CL final score for difficult tasks did not significantly decrease compared to the CL Initial score, and we observed numerous cases where performance on past tasks actually increased through BWT. Secondly, our model demonstrated higher performance compared to the baseline of existing CL methodologies.
>
> From these points, we can conclude that our model remains valid even for tasks with a high hierarchy where there are limitations in learning just a single task.
>
> Furthermore, to enhance our experimental evaluation, we have paid attention to the phenomenon related to Joint training shown in the results table. These inspection of our result is in **Experiment 2** of the global comment and **Fig.R1** and **R2** of the **PDF**.
> ***
> **W2** We agree with the comment. In the final version, we will include the accuracy of conventional CL that was compared in the supplementary section in the main paper.
> ***
> **W3** Following your comment, we conducted three repeated experiments for all settings by adding two additional seeds.
> |% std|RNN|Stack-RNN|Tape-RNN|LSTM|
> |---|---|---|---|---|
> |Joint|1.14|5.01|6.77|1.76|
> |Initia|1.55|1.41| 3.91|1.41|
> |Final|1.40|2.25|5.01|2.25|
> * Standard deviation from three separate runs of high-correlation CL-AR scenario (Table 3 main)
>
> With the exception of Tape-RNN, which is largely contingent on the initialization, minimal variance was observed, confirming the consistency of the content of this paper. In the final version table, we promise to include the results and variances from the repeated experiments.
> ***
> ## Questions
> Q1. CLeAR is a specialized model for AR tasks, serving a distinct purpose from LwF. Unlike images, where input distributions share common attributes like color, texture, and patterns, AR tasks present highly distinct input distributions unique to each task. Given the considerable dissimilarity between tasks and their varying dimensions, we introduce a novel approach called CLeAR to effectively adapt to these unique scenarios.
>
> The mapping function, first part of CLeAR, is designed to explicitly align different task inputs. We propose a novel approach to create mapping that can align distributions without prior knowledge of each task adopting Auto-Encoder architecture. This resulted in highly aligned distribution of all tasks (Fig.4-b in main).
>
> Otherwise, LwF assumes image distributions are similar across tasks. While this imperfect assumption leads to outdated performance in the image domain, it fortunately benefited by our mapping function. However, due to the regularization term in LwF that accounts for image distribution shift, LwF does not fully utilize the highly aligned mapping space, resulting in notably lower performance for ten complex tasks (Supp. table 5). On the other hand, our CLeAR consistently exhibits high performance for AR tasks, thanks to its AR specialized label sharpening term.
>
> Table 5 in appendix D shows that CLeAR is more capable of preserving information for long sequential CL. Especially for RNN. Additionally, we demonstrated that our CLeAR methodology surpasses not only LwF but also LwF w/ our mapping.
>
> ||CLeAR|||LwF (our mapping)|
> |---|---|---|---|---|
> |High-correlation|final(%)|forgetting(%)|final(%)|forgetting(%)|
> |MA| 60.09% |3.07| 55.37%|43.64%|
> |CN| 87.23% |0.46%| 82.95%|11.20%|
> |RS | 46.29% |1.52%| 50.30% |2.72%|
> |BS| 79.17% |2.73%| 75.31% |4.41%|
> * MA: modulus from 2 to 8
> * CN: cycle length from 2 to 8
> * RS: string alphabet count from 2 to 8
> * BS: string alphabet count from 2 to 8
>
>
> ||CLeAR|||LwF(our mapping)|
> |---|---|---|---|---|
> | Inter-hierarchy|final(%)|forgetting(%)|final(%)|forgetting(%)|
> |Ascending|60.22|7.55|59.42|12.07|
> |Descending|63.32|2.90|63.05|8.49|
>
> * 10 tasks of EP-CN-PC-CO-RS-SM-IP-OF-BS-DS. Ascending and descending order of hierarchy
>
> Experiments showed that the CLeAR is capable of long and challenging scenario and outperforms LwF (+our mapping)
> ***
> ## Limitations
> **L1** Thank you for the valuable feedback. We will cite two categories of AR-related papers and add relevant content to the related work section. We include papers such as DeepProbLog (2018), HOUDINI (2018), NeuralTerpret (2017), which tackle AR tasks using logic solvers, as well as Recognizing Long Grammatical Sequences Using Recurrent Networks Augmented With an External Differentiable Stack (2021 PMLR) and Learning Hierarchical Structures with Differentiable Nondeterministic Stacks (2022 ICLR), which attempted AR solutions using neural networks. Additionally, we will add ref [24] of the main paper to the related work section.
> ***
> **L2** We apologize for the inadequate explanation of the details of the benchmarks. In the final version, we will strengthen the explanation in the supplementary section and promise to publicly provide a comprehensive disclosure of the complete code we used and the precise experimental setups for each task.
> ***
> *p.s. Due to space constraints, we'll promptly provide the full table for W3, Q1 during the interactive review on request*

---

> > ### Author Response · Authors · 2023-08-18
> > **Looking forward to your post-rebuttal feedback**
> >
> > Dear Reviewer iTug
> >
> > Thank you again for the insightful comments and suggestions. Since the deadline for our discussion is approaching, we sincerely look forward to your follow-up response. We are happy to provide any additional clarification that you may need.
> >
> > For your convenience, we provide a summary below:
> >
> > * #### Strengthened experimental explanation and showed evidence for the strong performance of CLeAR across tasks of various difficulties (more details in **iPYe W1**).
> >
> > * #### Repeated multiple experiments with different seeds show the stability of results and we also reported standard deviation.
> >
> > * #### Motivation behind our CLeAR algorithm in the context of CL-AR, as well as the significant differences against LwF. This includes various additional comparative experiments.
> >
> > * #### Numerous related works concerning AR tasks have been added to the paper.
> >
> > We hope that the provided new experiments and the additional discussion have convinced you of the merits of this paper. Please do not hesitate to contact us if there are additional questions.
> >
> > Meanwhile, we thank you for your very helpful comments. It would indeed make our paper clearer and stronger.
> >
> > Thank you for your time and effort.
> >
> > Best regards, Authors

---

> ### Author Response · Authors · 2023-08-21
> **A kind reminder**
>
> ### Dear reviewer iTug
>
> We wanted to kindly remind you that the interactive discussion phase will be ending in just a few hours. Unfortunately, we won't be able to engage in further discussions with you after the deadline. We hope that our response has addressed your concerns, and turned your assessment to a more positive side. Please let us know if there are any other things that we need to clarify.
>
> We thank you so much for your helpful and insightful suggestion.
>
> Best, Authors

---

### Official Review · Reviewer_FdFh · 2023-07-10

**Soundness:** 3 good
**Presentation:** 3 good
**Contribution:** 2 fair
**Rating:** 4
**Confidence:** 3

**Summary:**

The paper looks at the continual learning (CL) setting involving tasks which require more abstract reasoning (reusable across input domains), s.a. addition and multiplication. The authors imagine that the inputs of all encountered tasks can be mapped into a common space, which, in turn, can be similarly transformed across tasks, in order to achieve transfer on a higher level. To take advantage of this fact, the inputs are processed by three NNs: 1) task-specific m_t which maps the inputs to a common space; 2) f which processes the resulting latent embeddings; 3) h_t which is a task-specific single-layer projection head. m_t is trained using an additional decoder. f and h_t are trained using a similar approach to Learning without Forgetting (LwF). The paper introduces a new set of tasks for evaluating abstract reasoning in a CL setting. The results demonstrate that this approach is capable of backward transfer on the created tasks.

**Strengths:**

The paper looks towards an interesting direction in CL. I found the discussion on the challenges around using most CL methods for abstract reasoning (AR) to be interesting.

The paper introduces a novel set of logical reasoning tasks which are categorised according to a class in the Chomsky hierarchy.

The experiments demonstrate the method’s capability of achieving backward transfer.

The use of the auto-encoder objective to learn a common mapping between tasks appears novel.


**Weaknesses:**

The method claims to introduce the first CL scenario for abstract reasoning, but simpler AR tasks have been explored in NeuralTerpret, HOUDINI, DeepProbLog.

The description of the algorithm seems rushed. The approach is presented but not well motivated, e.g. the properties of the mapping m_t on lines 218-222. The continual learning approach’s novelty seems limited, being mainly modelled after LwF.

My impression is that the method is not evaluated on a difficult enough setting in order to determine whether it is promising. Each input consists of one-hot encodings (as opposed to images) which I imagine are easier to learn how to map to a common space.


**Questions:**

What do you think would happen if two tasks have partially overlapping input alphabets, e.g. (task1: 1 2 3 4 , task2:  3 4 5)? Would this idea of mapping both input distributions to the same space be problematic?

What would you say is learned in f_theta across disparate tasks, which allows for transfer between them?


**Limitations:**

The fixed-size model limits the effective number of tasks which can be solved.

It appears that the approach relies on some level of similarity between tasks, as it uses the new inputs to create pseudo-labels.

---

> ### Author Rebuttal · Authors · 2023-08-09
>
> *We thank you for the insightful comments and suggestions. We have addressed each of your questions below.*
> ## Weaknesses
> **W1**
> Thank you for introducing us to these inspiring works. We will add this research to the related work section (we reported a summary in global comment). While all of these studies are remarkable, we think there are significant differences between their works and our work in terms of scenarios and methods.
>
> First, our work does not rely on any human (prior) knowledge about the tasks in its model construction. The model purely performs "Learning from data" without knowing incoming logical tasks. In contrast, their works require pre-configured programs that handle logic with human knowledge of future tasks. This contradicts the assumption of CL, as it is impossible to anticipate and program for all potential problems that may arise.
>
> Second, our paper performs logical operations using a pure neural network. While their works do utilize neural network structures internally, they mainly use them as feature extractors, such as extracting features from MNIST digits. Also for the lifelong learning scenarios they presented (e.g., Summing two images to perform arithmetic operations on multiple images), the neural network acts as a feature extractor, while pre-defined programs perform the actual logicals.
>
> Third, while our methodology allows CL, their works lack scalability to multiple tasks. They either modularize programs for each task or create parameterized programs from the beginning to accommodate multiple tasks.
> * * *
> **W2**
> We apologize for the unclear explanation of algorithm. In the final version, we will revise the description of our CLeAR methodology, motivated by the unique characteristics of AR tasks.
>
> CLeAR is a specialized model for AR tasks, serving a distinct purpose from LwF. Unlike images, where input distributions share common attributes like color, texture, and patterns, AR tasks present highly distinct input distributions unique to each task. Given the considerable dissimilarity between tasks and their varying dimensions, we introduce a novel approach CLeAR to effectively adapt to these environments.
>
> The mapping function, first part of CLeAR, is designed to explicitly align different task inputs. We propose a novel approach that can align distributions without prior knowledge of each task adopting Auto-Encoder architecture.
> Otherwise, LwF assumes image distributions are similar across tasks. While this imperfect assumption leads to outdated performance in the image domain, it fortunately benefited by our mapping function. However, due to the regularization term in LwF that accounts for image distribution shift, LwF does not fully utilize the aligned mapping, resulting in lower performance for ten complex tasks (W3). On the other hand, our CLeAR consistently exhibits high performance for AR tasks, thanks to its AR specialized label sharpening term.
> * * *
> **W3** We conducted additional experiments on more complex sequential tasks involving 10 different AR tasks (EP-CN-PC-CO-RS-SM-IP-OF-BS-DS: reported average).
> 1. Increasing hierarchy
> > CLeAR final 60.22%, BWT -7.55%
> LwF (our mapping) final 59.42%, BWT -12.07%
> 2. Decreasing hierarchy
> > CLeAR final 63.32%, BWT -2.90%
> LwF (our mapping) final 63.05%, BWT -8.49%
>
> Our model demonstrated favorable performance and less forgetting in these difficult scenarios.
> According to recent research (main[11]), even for tasks that may seem very simple to humans, like the parity-check task (whether the sum of binary is even or odd), transformer models completely fail to learn the algorithm. Therefore, We find it difficult to consider our task as a very easy task from the model's perspective.
> ## Questions
> **Q1** Thank you for suggesting good experiments. We conducted experiments on tasks that have semantically identical partially overlapping input alphabets.
>
> MA (modular arithmetic) task with fixed modulus 8, and for each task, input digit is (0,1,2,3), (2,3,4), (3,4,5), (4,5,6,7), (1,3,5,7), and (2,4,6,8) sequentially (reported average).
> > CLeAR final 66.17%, BWT 0.62%
> LwF (our mapping) final 60.98%, BWT -3.68%
>
> BS (bucket sort) task, same set of numbers sequentially included in the input.
> > CLeAR final 45.03%, BWT -1.85%
> LwF (our mapping) final 45.95%, BWT -0.55%
>
> CLeAR demonstrates favorable performance on MA-partial overlap achieving best Acc 99.81% with LSTM; however for BS-partial overlap, there are plenty of rooms to be improved.
> ***
> **Q2** To address your comment, we performed additional experiments. We observed and analyzed, for the first time, how a model learns certain AR tasks internally during a CL process and coordinates knowledge. Additionally, through this rebuttal, we made a notable observation that CL on AR can often outperform joint training. Based on these, we proposed hypotheses to explain this phenomenon. Please refer to Fig.R1 and R2 in the attached PDF file and iPYe Q1 for details.
> ## Limitations
> **L1** Your observation is valid, and indeed, increasing the model's capacity is a viable approach in the field of CL. There are foundational models like Progressive Neural Networks (2016) and Dynamically Expandable Networks (ICLR 2018). However, recent CL research has primarily been oriented toward using a fixed-sized neural network that is more challenging and resembles human learning processes.
> ***
> **L2** We agree with your insight. In fact, methodologies in CL aim to identify similarities between tasks and leverage them to facilitate the learning of subsequent tasks. Traditional CL on images has shown high similarity because all tasks involved recognizing general features like colors or stripes, which require similar kernels.
> However, in CL-AR, the distributions between such tasks vary significantly, making it challenging for the model to capture similarity as mentioned in W2.
>
> *Due to text limit, we'll promptly provide the full table in W3, Q1 during the interactive review on request.*

---

> > ### Author Response · Authors · 2023-08-18
> > **Looking forward to your post-rebuttal feedback**
> >
> > Dear Reviewer FdFh
> >
> > Thank you again for the insightful comments and suggestions. Since the deadline for our discussion is approaching, we sincerely look forward to your follow-up response. We are happy to provide any additional clarification that you may need.
> >
> > For your convenience, we provide a summary below:
> >
> > * #### Fundamental differences between the previous neuro-symbolic framework with NNs and our CLeAR, and critical reasons why only our approach is suited for CL.
> >
> > * #### Detailed motivation behind our CLeAR algorithm highlighted the key distinctions against LwF and we conducted additional comparative experiments (more details in **iTug Q1**)
> >
> > * #### Difficulty of AR tasks for NNs (even large transformers fail at simple (*to human*) PC task learning).
> >
> > * #### Additional experiments of lengthy and challenging **10-consecutive AR tasks** showing CLeAR exhibits minimal forgetting.
> >
> > * #### Additional experiments of **tasks of columns partially overlap** showing CLeAR could remember almost all tasks perfectly (99.8%).
> > * #### Newly devise a novel approach based on CL-AR to analyze how the model learns and transfers tasks in CL (details in the attached **PDF & iPYe Q1**)
> > * #### Discussions of the current trends in CL research where model size is either fixed or increased and methods utilizing the similarity between tasks for knowledge transfer.
> >
> > We hope that the provided new experiments and the additional discussion have convinced you of the merits of this paper. Please do not hesitate to contact us if there are additional questions.
> >
> > Meanwhile, we thank you for your very helpful comments. It would indeed make our paper clearer and stronger.
> >
> > Thank you for your time and effort.
> >
> > Best regards, Authors

---

> > ### Comment · Reviewer_FdFh · 2023-08-18
> >
> > Thank you for your reply.
> >
> > W2: Your reply does not alleviate my concern regarding the novelty of the approach. As f and h_t are trained using an objective similar to LwF, the novelty in the approach appears to be in: a) having a mapping m_t which maps to a common space, as well as in b) how m_t is trained. The idea of a) is already found in papers doing Domain Adaptation, leaving me to conclude that the novelty of the approach hinges on b). Could you explain what makes the way you train m_t novel? If this is the main contribution, shouldn't it be described more clearly in the paper (instead of being left to the Appendix)?
> >
> > W3: Thank you for providing the additional results. Could you clarify what these 10 AR tasks are? What are their inputs, what are the labels? My original concern was that it was perhaps not really challenging for the inputs used originally to be mapped to the same common space.
> >
> > Q1: Thank you for providing these extra results. Conceptually, the requirement that all mapping needs to span all of the common space, appears limiting in experiments with partial overlap. Moreover, what if the 2 tasks are too distinct and there's no overlap, thus mapping their inputs to the same common space and processing these with the same f might reduce the performance. In contrast, CL methods used for image processing of different domains do not have a penalty which forces the latent spaces between different domains to overlap. Would you agree that there's an inherent limitation as to the variety of the input domains functions which can be used?

---

> > > ### Author Response · Authors · 2023-08-19
> > > **Response to W2**
> > >
> > > Thank you very much for your kind reply. We have provided answers to your three questions sequentially.
> > >
> > > ## Main answer W2
> > > * **Regarding a)**, we believe that the alignment method  in domain adaptation differs both in its purpose and method from our mapping. The concept of domain variant, or task irrelevant information, does not apply to the CL-AR task. As the domain of images transitions from natural to cartoon, domain invariant (task relevant) information remains preserved, compensating for the altered domain variant features through alignment.
> > >
> > > * In contrast, in the AR task, all inputs are task relevant, and even a minor change in a single input pixel leads to a change in the output class. In essence, our mapping function represents a unique **one-to-many transformation**, which ensures a **uniform distribution** and a **function range equal to the codomain**, rather than preserving feature distribution. Furthermore, mapping retains every piece of information allowing a complete reconstruction of the original raw input through reverse mapping.
> > > ***
> > > * **Regarding b)**, the novelty of m_t stems from its pioneering capacity to enable CL for a sequence of arbitrary AR tasks. Furthermore, it has demonstrated comparable results, and in some cases even superior results, in a single training setting contrasted with using raw input, proving m_t does not hinder overall model’s performance.
> > >
> > > * The **novelty of the m_t learning method** lies in incorporating a perfect reconstruction loss to preserve the information from raw data and utilizing mean and covariance losses to transform each task into a discrete uniform distribution, ensuring that information is not lost in the process and allowing **one-to-many mapping**. This allowed CLeAR to perform well in adaptation to the CL-AR task, which was introduced for the first time. We shall describe this method more clearly in the final version.
> > > ***
> > > * **Our main contribution** is twofold: Firstly, we introduce a Continual Learning scenario for AR tasks, where neural networks learn logic continuously—a feat that was previously unattainable within neuro-symbolic frameworks. Secondly this pioneering work includes the proposition of the first baseline model, CLeAR, which makes this continuous learning of logic not only feasible but also performs high accuracy and minimal forgetting in various scenarios.
> > >
> > > * The CL-AR task possesses distinct data characteristics that pose challenges to applying existing methodologies used in CL for images or NLP. Furthermore, there is a scarcity of research on NN's AR tasks themselves. The newly proposed experiment of inspecting sequential acquisition of automata within its structure, marks a significant advancement. Consequently, this paper is poised to serve as a starting point for the novel field of CL-AR, providing valuable insights for future exploration.
> > > ***
> > > ## Appendix
> > > We have explored various methodologies in **domain adaptation**:
> > > In the field of Statistic Divergence Alignment, the focus is on minimizing domain discrepancy in the latent feature space. Methods such as MMD ([1] 2018 TPAMI) and its variants ([2] 2017 PMLR), CORAL estimation using inter-domain covariance ([3] 2020 AAAI), and methods utilizing Wasserstein distance ([4] 2020 AAAI; [5] 2021 ICASSP) measure the distance between features in each domain. These methods align these features to retain only the invariant feature. Approaches such as [6] (2020 arXiv) and OSUDA ([7] 2021, MICCAI) use batch statistics for distribution alignment.
> > > Field of generative domain mapping, similar to ours, change data to the target domain at the input data level but employ completely distinct methods, such as the style GAN ([8] 2020 TMI, [9] 2022 SPIE).
> > > Lastly, for robust representation learning, there are methodologies of pre-text tasks or contrastive learning ([10] 2019 IEEE Access, [11] 2020 arXiv). However, these self-supervised training methods are not feasible for AR tasks.
> > >
> > > [1] Rozantsev, et al. Beyond sharing weights for deep domain adaptation
> > > [2] Long, et al. Deep transfer learning with joint adaptation networks
> > > [3] Chen, et al. Homm: Higher-order moment matching for unsupervised domain adaptation
> > > [4] Liu, et al. Importance-aware semantic segmentation in self-driving with discrete wasserstein training
> > > [5] Ge, et al. Embedding semantic hierarchy in discrete optimal transport for risk minimization
> > > [6] Zhang, et al. Generalizable semantic segmentation via model-agnostic learning and target-specific normalization
> > > [7] Liu, et al. Adapting off-the-shelf source segmenter for target medical image segmentation
> > > [8] Yang, et al. Unsupervised MR-to-CT synthesis using structure-constrained CycleGAN
> > > [9] Liu, et al. Structure-aware unsupervised tagged-to-cine MRI synthesis with self disentanglement
> > > [10] Xu, et al. Self-supervised domain adaptation for computer vision tasks
> > > [11] Kim, et al. Cross-domain self-supervised learning for domain adaptation with few source labels

---

> > > ### Author Response · Authors · 2023-08-19
> > > **Response to W3**
> > >
> > > ## Main answer W3
> > > * **CL on AR tasks with diverse column numbers was immensely challenging.** We attempted techniques like padding, MLP & attention based embedding, and pre-trained language model-based embeddings (sota in tabular transfer learning), yet all coupled with conventional CL methods resulted in complete failure.
> > >
> > > * However, our newly devised mapping function, m_t, has led to the creation of a remarkably high-performing and stable model for CL-AR, despite its **lightweight structure and intuitive training approach**. Unlike existing input space matching methods that struggled, our m_t approach has demonstrated superior performance and stability.
> > >
> > > * We haven't pushed the limits of our mapping function's performance by increasing the number of alphabets extensively. This decision was made to ensure a **fair comparison with recent papers ([1] 2022 ICLR, [2] 2022 ICLR)** that experimented with a small number of alphabets for single AR task performance evaluation. Also the main goal was to determine whether CLeAR can handle a multiple sequence of consecutive AR tasks. We have shown that single-task performance remains comparable to these papers with our mapping function , validating that our mapping does not hinder performance.
> > >
> > > * As you've suggested, **expanding the alphabet's range and testing the mapping function's performance** to its limits for both single tasks and CL could be a valuable future direction. It would help determine if models thought to handle specific AR tasks are influenced by the alphabet's count. Additionally,  it will offer the opportunity to ascertain whether models traditionally believed capable of handling specific AR tasks remain unaffected by the alphabet's count.
> > > ***
> > > * **The most challenging task for the mapping function**, among our experiments, was the high-correlation CL-AR scenario on modular arithmetic (**mod 8**).
> > > In the MA task, the input alphabet consists of **eight digits 0 to 7** and __three operators +, -,*.__  The output corresponds to numbers ranging from 0 to 7, obtained through arithmetic operations. In this setup, we **changed the modulus of this task from 2 to 8**, creating a sequence of seven tasks. For instance, in the input sequence, digits would be arranged as 0,1 / 0,1,2 / … / 0-7 for each task. Each task's mapping function learned to handle the mapping of these varying numbers of alphabets.
> > > Despite this complexity, CLeAR demonstrated excellent performance in this scenario as well (refer to the main paper Table 2: achieving the best accuracy of 91.6% with LSTM).
> > > ***
> > > ## Appendix
> > > 10 AR tasks are as follows: **EP-CN-PC-CO-RS-SM-IP-OF-BS-DS**
> > > * **Regular language** (Finite automata) should learn “internal state”
> > > 1. **EP; Even paires**:  input **0,1**(binary) of arbitrary length(1~100) / output: 0,1
> > > Check if the count of 01 and 10 is the same among the pairs of two consecutive numbers
> > > 2. **CN; Cyclic navigation**: input **move left, stay, move right** of arbitrary length(1~100) /  output: final cycle position (0,1,2,3,4)
> > > The position after executing a command on a cycle with 5 positions.
> > > 3. **PC; Parity Check**: input **0,1** (binary) of arbitrary length(1~100)/ output: 0,1
> > > whether the sum of the sequence is odd or even
> > > * **Context free language** (Push down automata) should learn “stack memory”
> > > 4. **CO; Compare Occurrence**: input **0,1,2,3** of arbitrary length(1~100)/ output: 0,1,2,3
> > > Return the digit that appears most frequently in the string.
> > > 5. **RS; Reverse String**: input **0,1** (binary) of arbitrary length(1~100)/ output: reverse sequence (input length)
> > > 6. **SM; Stack Manipulation**: input: **digit 0, digit 1, action pop, action push 0, action push 1**/ output: string with 0,1
> > >  return string after following instruction
> > > * **Context sensitive language** (Linear bounded automata) should learn “tape memory”
> > > 7. **IP; Interlocked Pairing**: input **n*0s m*1s** (total length 1~100) / output n*0s (n+m)*1s m*0s (total length 2*input length)
> > > 8. **OF; Odds First**: input **0,1** (binary) of arbitrary length(1~100)/ output: reorder string to digits in odd index comes first (input length)
> > > 9. **BS; Bucket Sort**: input **0,1,2,3** of arbitrary length(1~100)/ output: input string sorted in ascending order (input length)
> > > 10. **DS; Duplicate String**: input **0,1** (binary) of arbitrary length(1~100)/ output: input string repeated twice (2*input length)
> > > *Please refer Appendix C.2 for more detail*
> > >
> > > [1] Delétang, et al. Neural networks and the chomsky hierarchy
> > > [2] Liu, et al. Transformers learn shortcuts to automata

---

> > > ### Author Response · Authors · 2023-08-19
> > > **Response to Q1**
> > >
> > > ## Main answer Q1
> > > * Our experiments highlighted that our model **exhibited minimal forgetting and knowledge transfer** with **partially overlapping columns performing the same role within the same task** (this additional experiment), **overlapping columns with the same alphabet in similar tasks** (high-correlation CL-AR scenario), **tasks with distinct alphabets and no overlap** (in-hierarchy CL-AR scenario), and even when dealing with **tasks requiring different automata hierarchies** (inter-hierarchy CL-AR scenario). Furthermore, additional experimentation (iPYe Q1) demonstrated that this strong performance might stem from the internal learning of automata and aligning shareable states across tasks.
> > >
> > > * Our mapping also differs in purpose and usage from the feature alignment typically used in **conventional domain CL methods**, and it is applicable to most AR tasks expressed as formal languages.
> > > Traditional CL methods used for image processing across different domains have employed techniques such as introducing domain adaptation or co-training from diverse domains. For instance, in [1] (2021 AAAI), the model learns simultaneously from images of distinct domains and updates weights in the direction of gradient vectors with positive inner products. [2] (2022 CVPR) employs a Mahalanobis similarity matrix to align different domains while preserving the manifold by expanding or shrinking each axis of feature. [3] (2022 CVPR) trains a student model using the weights of a pretrained model and the moving average values. In [4] (2023 CVPR), few specific weights are updated for new domain adaptation.
> > >
> > > * These methods all rely on the fact that, even though the domains differ, there exists a strong similarity among images of the same class. For instance, even if a natural image of a car and a cartoon image of a car belong to different domains, they share characteristic shapes and textures. Therefore, feature alignment in domain continual learning penalizes the feature manifold slightly to align features across domains, as exemplified by [2] (2022 CVPR).
> > >
> > > * **In contrast, our mapping space is distinct from the aligned feature space.** In AR tasks, the input distribution consists largely of white noise rather than a thin manifold of meaningful images. While domain adaptation aims to preserve and align the feature manifold, our mapping function's objective is to transform all input distributions into the same distribution. As a result, both the objective and the approach differ. Thus, considering CL-AR tasks as domain adaptation is inappropriate in my view. Furthermore, for AR tasks, our mapping can effectively transform various formal languages into a uniform distribution.
> > > ***
> > > ## Appendix
> > > [1] Tang, et al. Gradient regularized contrastive learning for continual domain adaptation
> > > [2] Simon, et al. On generalizing beyond domains in cross-domain continual learning
> > > [3] Wang, et al. Continual test-time domain adaptation
> > > [4] Prasanna, et al. Continual Domain Adaptation through Pruning-aided Domain-specific Weight Modulation
> > > ***
> > > We once again appreciate your insightful comments.

---

> > > ### Author Response · Authors · 2023-08-21
> > > **A kind reminder**
> > >
> > > ### Dear reviewer FdFh
> > >
> > > We wanted to kindly remind you that the interactive discussion phase will be ending in just a few hours. Unfortunately, we won't be able to engage in further discussions with you after the deadline. We hope that our response has addressed your concerns, and turned your assessment to a more positive side. Please let us know if there are any other things that we need to clarify.
> > >
> > > We thank you so much for your helpful and insightful suggestion.
> > >
> > > Best, Authors

---

### Author Rebuttal · Authors · 2023-08-09

We thank all the reviewers very much for their valuable comments and constructive suggestions to strengthen our work. Also for the positive comments and encouraging remarks: The paper addresses for the first time the continual learning (CL) adaptation of algorithmic reasoning(AR) tasks (iTug, iPYe). It also introduces a novel set of logical reasoning tasks for CL which are categorized according to the Chomsky hierarchy (FdFh, aCtv). These are disparate from the standard CL scenario (iTug) and research has not yet been conducted (GYfA). Also, it is challenging to use most previous CL methodologies on this new task (FdFh). The paper introduces a novel methodology, CLeAR (GYfA), and the core idea of mapping tasks and inputs to an embedding space from which a unified model learns continually to solve tasks is, to my knowledge, highly novel (iPYe), novel (FdFh), makes sense (aCtv). With several experiments, the proposed method showed desired effects, preventing catastrophic forgetting in several scenarios (iTug) and showing the capability of achieving backward transfer (FdFh). The paper bridges the gap between CL and real-world cognitive skills development (GyfA) and leads to a more general setting w.r.t. standard CL which can improve current research and lead to new methodologies (iTug).

Following reviewers’ suggestions, we have added literature reviews and performed more supporting experiments. Here, we would like to highlight the **main revisions**:

# Literature reviews
We cited and discussed the following papers
> **DeepProbLog(2018)[1], HOUDINI(2018)[2], NeuralTerpret(2017)[3]**: These papers propose a neuro-symbolic or probabilistic framework for solving logical problems. These frameworks are divided into two main parts. The first part consists of task-specific pre-defined parameterized functions capable of handling logic (e.g., ProbLog for DeepProbLog, symbolic program synthesizer for HOUNDINI, and program interpreter for NeuralTerpret). The second part involves Neural networks (NN). These frameworks support end-to-end gradient-based updates for the combination of the parameterized functions and the NN. The program part acts as a logic solver while NN acts as a feature extractor from the image. We included this paper In the "related work" section and described their relevance and differences compared to our research (by FdFh, GYfA).

> **On warm-starting neural network training[4], Does the Adam Optimizer Exacerbate Catastrophic Forgetting?[5]**: This paper discusses catastrophic forgetting, one of the core challenges in CL. It addresses the issue of rapid forgetting by preventing abrupt changes in model parameters. We have added this paper to the "introduction" and revised its relevant content (by aCtv).

> **A formal hierarchy of RNN architectures[6], Saturated transformers are constant-depth threshold circuits[7]**: These papers discuss the feasibility of initial AutoRegressive (AR) tasks on RNN and Transformer model architectures. We have added citations to these papers in the “related work” section (by aCtv).

# Experiments
> **Repeated experiments**: We updated single-run experiments to three repeated experiments by adding two more seeds. We confirmed that there were only minor variations in the values of the experimental results. In the final version, we will conduct many more repeated experiments, and this measure will further reduce confirmation bias (by iTug. refer **iTug W3** for more detail).

> **In-depth inspection of the model learning reasoning**: “What is learned in the model across disparate tasks, allowing knowledge transfer.”(FdFh), “How does the model coordinate incoming tasks?”(iPYe), “Is it really good to learn AR sequentially instead of learning all at once?”(aCtv) These are very important questions in CL, but it has been a challenging problem due to the black-box nature of the NN. However, we have taken a step forward in solving these problems with the help of the high interpretability of the AR task and automata theory. We conducted an analysis of **A. what the model actually learns from data and how they coordinate it with the new task** by analyzing the model’s hidden feature with the corresponding automata. And **B. what differences are achieved between CL and joint training.** In typical CL scenarios, joint training (all tasks learned together), is often considered an upper bound in terms of performance. However, contrary to the conventional belief, joint training often showed lower performance in the conducted AR tasks, which emphasizes the importance of continually learning AR tasks just like humans improve their logical abilities through step-by-step learning in the real world. We quantitatively compared the performance of joint training and CL and proposed a convincing hypothesis for why CL outperforms joint training in certain cases (by FdFh, iPYe. For the result of this experiment, refer **Fig R1,R2 of attached PDF** and **iPYe Q1** for more detail).

> **Experiments on more difficult tasks**: We extended our experiments to more model structures on the challenging ten sequential AR task scenarios from our paper, along with reversed order of the Chomsky hierarchy. Furthermore, we conducted additional experiments to investigate cases of tasks with partially overlapping columns (by FdFh. refer **FdFh W3,Q1** for more detail).

* The code for additional experiments will also be publicly released in the final version.

# References
[1] Manhaeve, et al. "Deepproblog: Neural probabilistic logic programming."
[2] Valkov, et al. "Houdini: Lifelong learning as program synthesis."
[3] Gaunt, et al. "Differentiable programs with neural libraries."
[4] Asht, et al. "On warm-starting neural network training."
[5] Ashley, et al. "Does the Adam Optimizer Exacerbate Catastrophic Forgetting?."
[6] Merrill, et al. "A formal hierarchy of RNN architectures."
[7] Merrill, et al. "Saturated transformers are constant-depth threshold circuits."

---

### Decision · Program_Chairs · 2023-09-21

**Decision:**

Accept (poster)

**Comment:**

This paper studies continual learning of algorithms for the first time. Authors introduce a novel set of logical reasoning tasks classified into Chomsky hierarchies. Then they propose a methodology that involves one-to-many mapping of input to a shared representation space. The experimental results demonstrate that their methodology (CLeAR) achieves zero forgetting and backward transfer. All reviewers except one voted for acceptance. After reading the reviews and authors detailed response, I am recommending  acceptance. Please take reviewers suggestions into account for the camera-ready version. In particular, I think it is important to discuss the motivation for this line of work in light of foundation models and how they relate.